# Reconstructing Antarctic winter sea-ice extent during Marine Isotope Stage 5e

Matthew Chadwick[1,2*]; Claire S. Allen[1]; Louise C. Sime[1]; Xavier Crosta[3] & Claus-Dieter Hillenbrand[1]

[1.] *British Antarctic Survey, High Cross, Madingley Road, Cambridge, CB3 0ET, UK*

[2.] *Ocean and Earth Science, National Oceanography Centre, University of Southampton Waterfront Campus, European Way, Southampton, SO14 3ZH, UK*

[3.] *Université de Bordeaux, CNRS, EPHE, UMR 5805 EPOC, Pessac, France*

*Corresponding author:* machad27@bas.ac.uk, British Antarctic Survey, High Cross, Madingley Road, Cambridge, UK

## Abstract

Environmental conditions during Marine Isotope Stage (MIS) 5e (130-116 ka) represent an important 'process analogue' for understanding the climatic responses to present and future anthropogenic warming. The response of Antarctic sea ice to global warming is particularly uncertain due to the short length of the observational record. Reconstructing Antarctic winter sea-ice extent during MIS 5e therefore provides insights into the temporal and spatial patterns of sea-ice change under warmer than present climate. This study presents new MIS 5e records from nine marine sediment cores located south of the Antarctic Polar Front, between 55 and 70 °S. Winter sea-ice extent and sea-surface temperatures are reconstructed using marine diatom assemblages and a Modern Analog Technique transfer function, and changes in these environmental variables between the three Southern Ocean sectors are investigated. The Atlantic and East Indian sector records show much more variable MIS 5e winter sea-ice extent and sea-surface temperatures than the Pacific sector records. High variability in the Atlantic sector winter sea-ice extent is attributed to high glacial meltwater flux in the Weddell Sea, indicated by increased abundances of the diatom species *Eucampia antarctica* and *Fragilariopsis cylindrus*. The high variability in the East Indian sector winter sea-ice extent is conversely believed to result from large latitudinal migrations of the flow bands of the Antarctic Circumpolar Current, inferred from latitudinal shifts in the sea-surface temperature isotherms. Overall, these findings suggest that Pacific sector winter sea ice displays a low sensitivity to warmer climates. The different variability and sensitivity of Antarctic winter sea-ice extent in the three Southern Ocean sectors during MIS 5e may have significant implications for the Southern Hemisphere climatic system under future warming.

## 1. Introduction

Antarctic sea ice is a critical part of the Southern Ocean (SO) and global climate system (Maksym, 2019). The vast extent of Antarctic sea ice and its huge seasonal variability (from ~4 x 10$^6$ km$^2$ in summer to ~18 x 10$^6$ km$^2$ in winter in the present day) have a strong albedo-radiation feedback (Hall, 2004). Brine rejection during sea-ice formation contributes to the production of dense shelf and bottom water masses, which, in turn, influence the strength of global overturning ocean circulation (Abernathey et al., 2016; Rintoul, 2018). Sea-ice cover also regulates heat and gas exchange between the SO and the atmosphere as well as phytoplankton productivity by acting as a physical barrier (Rysgaard et al., 2011) and barrier to sunlight and, when melting, causing stratification of the upper part of the water column (Goosse and Zunz, 2014).

Modern Antarctic sea-ice extent has shown a rapid decline since 2014 after four decades of gradual expansion (Parkinson, 2019). Within this overall trend there is substantial spatial heterogeneity in regional sea-ice trends, with decreases in the Bellingshausen and Amundsen seas concurrent with increases in the Weddell Sea and Ross Sea sectors (Hobbs et al., 2016; King, 2014; Parkinson, 2019). Alongside the inter-annual Antarctic sea-ice trends (Parkinson, 2019), there are also trends in seasonal variability, with the Amundsen Sea showing a substantial decrease in summer and autumn sea-ice concentrations but a slight increase in winter and spring sea-ice concentrations (Hobbs et al., 2016). Model simulations are unable to replicate the modern sea-ice changes without reduced regional warming trends (Rosenblum and Eisenman, 2017). Difficulties in reproducing modern sea-ice trends indicate the complexities of the climate dynamics that influence sea-ice extent in the SO today at different timescales (Ferreira et al., 2015; Hobbs et al., 2016; King, 2014; Purich et al., 2016; Stammerjohn et al., 2008).

Rising greenhouse gas concentrations are driving current global warming, with polar regions warming twice as fast (0.5 $^o$C per decade) as the global average (IPCC, 2019) and Antarctic winter sea-ice extent (WSIE) predicted to shrink by 24-34 % by C.E. 2100 (Meredith et al., 2019). However, the very short length of observational records in high latitudes together with the complexity of the climate system, as mentioned above, limit our understanding of the underlying processes and ability to accurately predict future changes. Past warm periods can help document the amplitude of sea-ice extent reduction and, therefore, help guide our understanding of the impacts of future climate change in polar regions.

Interglacial Marine Isotope Stage (MIS) 5e (130-116 ka; Lisiecki and Raymo (2005)) is the latest period when global mean annual atmospheric temperatures were warmer than present (~1 $^o$C; Fischer et al. (2018)) and global sea levels were higher than present (~6-9 m; Kopp et al. (2009)). Summer sea-

surface temperatures (SSSTs) in the SO peaked at an average of 1.6 ± 1.4 °C warmer than present at and north of the modern Antarctic Polar Front during this period (Capron et al., 2014; Shukla et al., 2021). MIS 5e warming is primarily orbitally forced, unlike current and future anthropogenic warming which is driven by rising greenhouse gas concentrations. Whilst MIS 5e cannot be considered a direct analogue for greenhouse gas induced global warming, it still represents an important 'process analogue' for understanding climate mechanisms and responses that are active under warmer-than-present climate conditions (Stone et al., 2016).

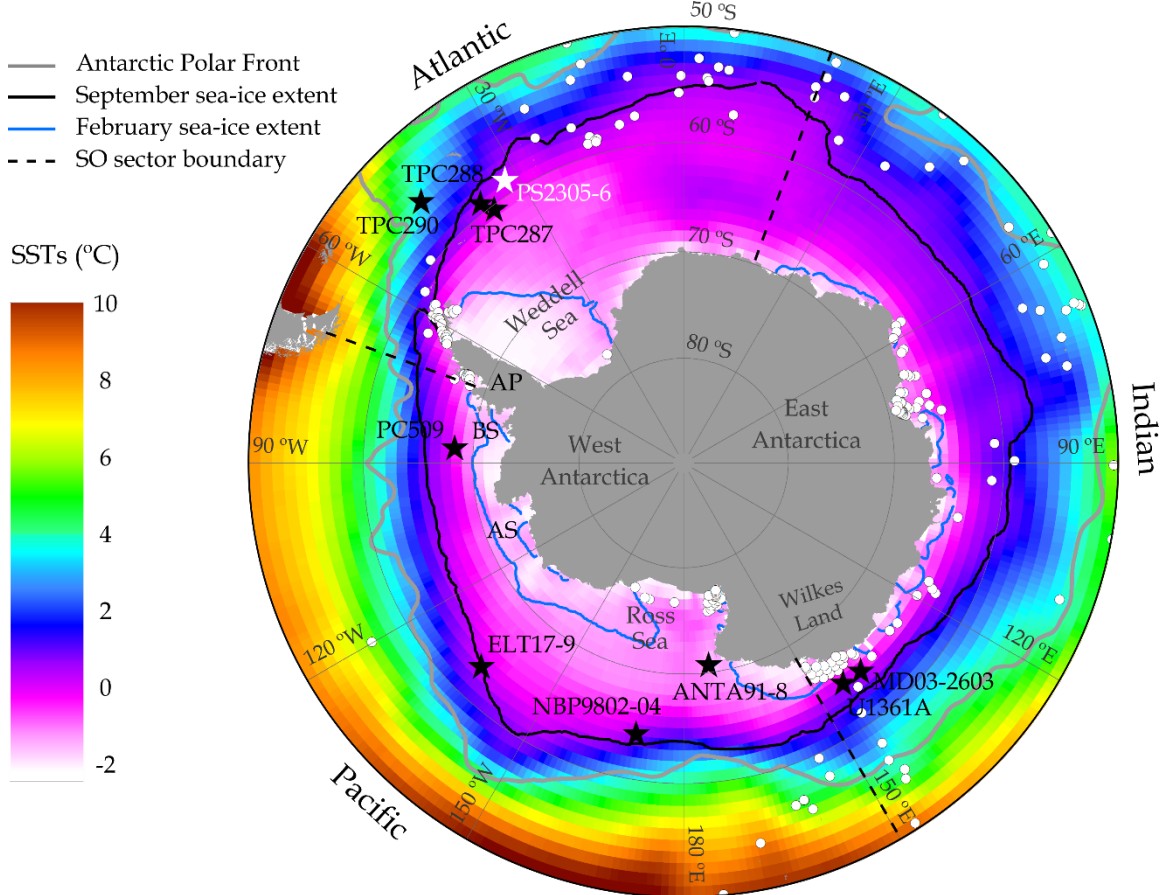

**Figure 1:** Map of core locations (black stars – this study, white star – Bianchi and Gersonde (2002)) with the modern (1981-2010) mean annual SSTs (COBE-SST2 dataset provided by the NOAA PSL, Boulder, Colorado, USA (https://psl.noaa.gov/)) and modern (1981-2010) median September and February sea-ice extents (data from Fetterer et al. (2017)). White dots mark the locations of surface sediment samples (located south of 50 °S) used as a modern reference dataset for the Modern Analog Technique transfer function. The black solid line is the September sea-ice extent (15 % cover) and the blue solid line is the February sea-ice extent (15 % cover). The grey solid line is the position of the modern Antarctic Polar Front (Trathan et al., 2000). The black dashed lines mark the boundaries between the three SO sectors (Atlantic, Indian and Pacific). AP – Antarctic Peninsula, BS – Bellingshausen Sea, AS – Amundsen Sea.

Diatoms preserved in SO marine sediments have been used for over 40 years to reconstruct past changes in Antarctic sea-ice extent and sea-surface temperatures (SSTs) (Armand and Leventer, 2010; Burckle et al., 1982; Thomas et al., 2019) due to the close relationship between their biogeographic

distribution patterns and surface water environmental conditions (Armand et al., 2005; Crosta et al.,
2005; Esper et al., 2010; Gersonde and Zielinski, 2000; Romero et al., 2005; Zielinski and Gersonde,
1997). Several previous studies have used model simulations, alongside limited data constraints from
marine sediment cores, to reconstruct SO WSIE and SSTs during MIS 5e (Capron et al., 2017; Holloway
et al., 2017; Holloway et al., 2018). However, there are currently no marine core records located far
enough south to constrain the predicted WSIE during MIS 5e (Chadwick et al., 2020; Holloway et al.,
2017). Due to chronological uncertainties in SO proxy records (Govin et al., 2015), previous studies
have assumed the minimum WSIE occurred synchronously around Antarctica and was coincident with
peak atmospheric temperatures in Antarctic ice cores at 128 ka (Holloway et al., 2017).
This study presents new reconstructions of SO winter sea ice (WSI) during MIS 5e from the diatom
assemblages preserved in nine marine sediment cores located south of 55 $^o$S and south of the modern
Antarctic Polar Front (Figure 1). Qualitative reconstructions are based on the occurrence of sea-ice
related diatoms (Gersonde and Zielinski, 2000). Quantitative estimates are produced through a
diatom-based Modern Analog Technique transfer function, based on numerous core-top sediment
samples (Figure 1) and originally detailed in Crosta et al. (1998). Quantitative and qualitative
reconstructions of WSIE in the three SO sectors; the Atlantic sector (70 $^o$W – 20 $^o$E), the Indian sector
(20 $^o$E – 150 $^o$E) and the Pacific sector (150 $^o$E – 70 $^o$W), are compared to answer the following
questions:
-  Did the minimum WSIE occur synchronously throughout the SO during MIS 5e?
-  Was the WSIE minimum concurrent with the peak Antarctic air temperatures at 128 ka?
-  Were the patterns in MIS 5e sea-ice change consistent between SO sectors?
**2.  Materials and methods**
*2.1.  Core sites*
The nine sediment cores used in this study are shown in Figure 1 alongside modern SSTs and sea-ice
extents. Details for each core are listed in Table 1. These cores were chosen as they contain >20 cm
thick intervals of diatom-rich sediments deposited during MIS 5e (including Termination II) and are
located further south than almost all previously published MIS 5e sea ice records (Chadwick et al.,
2020). Due to the locations of core sites MD03-2603 and U1361A, our SST and sea-ice reconstructions
for the Indian Ocean sector of the SO may reflect conditions only representative for the eastern Indian
sector.

### 2.2. Diatom counts

For the diatom assemblage data, microscope slides were produced using a method adapted from Scherer (1994). Samples of 7-28 mg were exposed to 10% Hydrochloric acid to remove any carbonate, 30% Hydrogen peroxide to break down organic material and a 4% Sodium Hexametaphosphate solution to promote disaggregation and placed in a warm water bath for a minimum of 12 hours. The material was homogenised, transferred into a ~10 cm high water column and allowed to settle randomly onto coverslips over a minimum of 4 hours. The water was drained away and coverslips were mounted on microscope slides with Norland Optical Adhesive (NOA 61). Slides were examined using a light microscope (Olympus BH-2 at x1000 magnification) and a minimum of 300 diatom valves were counted in each sample.

| Core | Latitude (º), Longitude (º) | Water depth (m) | Cruise, Year | Ship | Core length (cm) |
|---|---|---|---|---|---|
| TPC290 | -55.55, -45.02 | 3826 | JR48, 2000 | *RRS James Clark Ross* | 1179* |
| TPC288 | -59.14, -37.96 | 2864 | JR48, 2000 | *RRS James Clark Ross* | 940* |
| TPC287 | -60.31, -36.65 | 1998 | JR48, 2000 | *RRS James Clark Ross* | 615* |
| MD03-2603 | -64.28, 139.38 | 3320 | MD130, 2003 | *R/V Marion DuFresne II* | 3033 |
| U1361A | -64.41, 143.89 | 3459 | IODP Exp. 318, 2010 | *R/V JOIDES Resolution* | 38800 |
| ELT17-9 | -63.08, -135.12 | 4935 | ELT17, 1965 | *R/V Eltanin* | 2018 |
| NBP9802-04 | -64.20, -170.08 | 2696 | PA9802, 1998 | *R/V Nathaniel B. Palmer* | 740 |
| PC509 | -68.31, -86.03 | 3559 | JR179, 2008 | *RRS James Clark Ross* | 989 |
| ANTA91-8 | -70.78, 172.83 | 2383 | ANTA91, 1990 | *R/V Cariboo* | 511 |

**Table 1:** Details of the location and recovery information for the nine marine sediment cores analysed in this study. Cores are ordered by sector (Atlantic - East Indian - Pacific) and then latitude. * For each of the three TPC cores (TPC290, TPC288 and TPC287), the trigger core (TC) and piston core (PC) were spliced together to produce a composite record.

The combined relative abundance of *Fragilariopsis curta* and *F. cylindrus* (FCC) is used as a qualitative indicator of WSI presence (Gersonde and Zielinski, 2000), with abundances >3 % associated with locations south of the mean WSI edge, abundances 1-3 % found between the mean and maximum WSI edge and abundances <1 % indicative of conditions north of the maximum WSI edge (Gersonde et al., 2005; Gersonde and Zielinski, 2000). The relative abundance of the diatom species *Azpeitia tabularis* is used as a comparison with reconstructed SSSTs. *Azpeitia tabularis* is a warm water species restricted to the region north of the maximum WSIE (Zielinski and Gersonde, 1997), with abundances <5 % in surface sediments south of the modern Antarctic Polar Front (Esper et al., 2010; Romero et

al., 2005). Increasing abundances of this species in high latitude SO sediments therefore indicate
warmer SSTs and ice-free conditions.

### 2.3. *Modern Analog Technique (MAT)*

September sea-ice concentrations (SIC) and SSSTs (January to March) are estimated by applying a MAT
transfer function to the MIS 5e diatom assemblages. The MAT compares the relative abundances of
33 diatom species in each MIS 5e sample to the abundances of the same species in a modern reference
dataset composed of 257 surface sediment samples (modern analogs) from the SO. Modern
conditions for each surface sediment sample are interpolated on a $1^{\circ}$ x $1^{\circ}$ grid, with SSSTs from the
World Ocean Atlas 2013 (Locarnini et al., 2013) and September SIC from the numerical atlas of
Schweitzer (1995). The MAT was implemented using the "bioindic" R-package (Guiot and de Vernal,
2011), with chord distance used to select the 5 most similar modern analogs to each MIS 5e
assemblage. A cut-off threshold, above which any analogs are deemed too dissimilar to the MIS 5e
sample, is fixed as the first quartile of random distances determined by a Monte Carlo simulation of
the reference dataset (Simpson, 2007). The MAT257-33-5 (based on 257 reference samples, 33 taxa
and up to 5 analogs) utilised in this study is an evolution of the MAT195-33-5 detailed in Crosta et al.
(1998), with the addition of a further 62 surface sediment samples (Figure 1). The incremental
evolutions of this transfer function over the last 20 years have yielded robust SST and sea-ice
reconstructions when compared alongside other proxies within the same cores (e.g. Civel-Mazens et
al., 2021; Crosta et al., 2004; Ghadi et al., 2020; Nair et al., 2019; Shemesh et al., 2002).
Quantitative estimates of September SIC and SSSTs are produced for each MIS 5e sample from a
distance-weighted average of the climate values associated with the selected analogs. The
reconstructed SSSTs have a Root Mean Square Error of Prediction (RMSEP) of 1.09 $^{\circ}$C and an $R^2$ of
0.96, and the reconstructed September SIC have a RMSEP of 9 % and an $R^2$ of 0.93. The reconstructed
September SIC and SSST for each MIS 5e sample only use analogs below the dissimilarity threshold
and therefore could be reconstructed from less than 5 analogs in some samples. It is also possible to
get no-analog conditions, where none of the reference surface sediment samples are similar enough
to a MIS 5e sample, and it is therefore not possible to reconstruct September SIC and SSST for this MIS
5e sample.

### 2.4. *Diatom preservation*

For both the MAT and the FCC proxy, it is important that the diatom assemblage is well preserved, as
high dissolution causes preferential loss of the more lightly silicified diatom species, generally sea-ice
related species, and would therefore bias reconstructions towards warmer SSTs and lower sea-ice

conditions. The samples used in this study were investigated for signs of dissolution following the procedure detailed in Warnock et al. (2015), whereby the areolae in *F. kerguelensis* valves were checked to ensure there was little, or no, expansion and conjoining, as would occur under a high degree of dissolution. Diatom assemblages in the analysed samples were also checked for a mixture of both heavily and weakly silicified diatoms across the whole size range, which was suggested by Zielinski (1993) as an indicator of good preservation. Poor preservation of diatoms in sediments located beneath heavy winter sea ice (SIC >75 %) has likely limited most previous attempts to reconstruct MIS 5e conditions from core sites located south of the modern mean WSIE, and thus the preservation of samples analysed in this study was carefully considered to avoid introducing a warm (low sea ice) bias into our reconstructions.

| Core | SO sector | Chronology for MIS 5e | Chronological uncertainty (ka) |
|---|---|---|---|
| TPC290 | Atlantic | Correlating MS from TPC290 to EDC ice core dust record combined with *C. davisiana* abundances (Pugh et al., 2009)* | ± 2.6 |
| TPC288 | Atlantic | Correlating MS from TPC288 to EDC ice core dust record combined with *C. davisiana* abundances (Pugh et al., 2009) | ± 2.5 |
| TPC287 | Atlantic | Correlating MS from TPC287 to MS in core TPC288 (Chadwick et al., 2022) | ± 2.6-2.7 |
| MD03-2603 | East Indian | Correlating Ba/Al and Ba/Ti ratios from MD03-2603 to LR04 benthic oxygen isotope stack combined with diatom biostratigraphy (Presti et al., 2011) | ± 2.6 |
| U1361A | East Indian | Correlating Ba/Al ratios and lithological changes to the LR04 benthic oxygen isotope stack combined with LOD *H. karstenii* (Wilson et al., 2018) | ± 2.6-2.7 |
| ELT17-9 | Pacific | Combined abundance stratigraphies of *E. antarctica* and *C. davisiana* on SPECMAP age scale (Chase et al., 2003) | ± 2.5 |
| NBP9802-04 | Pacific | Correlating MS from NBP9802-04 to EDC ice core dust record combined with LOD *H. karstenii* (Williams, 2018) | ± 2.7 |
| PC509 | Pacific | Correlating wet bulk density (= proxy mirroring biogenic opal content) from PC509 to the LR04 benthic oxygen isotope stack (Chadwick et al., 2022) | ± 2.6-2.7 |
| ANTA91-8 | Pacific | Correlating MS from ANTA91-8 to the LR04 benthic oxygen isotope stack combined with LCO *Rouxia* spp. (this study; Figure 2) | ± 2.6 |

**Table 2:** Summary of the location and chronologies for the nine sediment cores analysed in this study. Cores are ordered by sector (Atlantic – East Indian - Pacific) and then latitude. LOD: Last Occurrence datum, LCO: Last Common Occurrence *For core TPC290 the chronology has been slightly adjusted from the published record of Pugh et al. (2009) by shifting the Termination II tiepoint to better align the magnetic susceptibility (MS) record with the dust record of the EPICA Dome C (EDC) ice core in East Antarctica (Chadwick et al., 2022).

## 3. Age models

### 3.1. Published chronologies

Eight of the sediment cores presented in this study have previously published age models, summarised in Table 2. Cores TPC290, TPC288, TPC287 and NBP9802-04 are published on the EDC3 chronology, cores MD03-2603, U1361A and PC509 are published on the LR04 chronology and core ELT17-9 is published on the SPECMAP chronology. These published chronologies are further constrained by checking the abundance of the diatom species *Rouxia leventerae* in all MIS 5e samples. All diatom assemblages analysed in this study have *R. leventerae* abundances <1 %, which suggest that the considered sediments are younger than the ~135 ka Last Occurrence Datum identified by Zielinski et al. (2002). To allow for consistent comparison of timings between cores, all cores are translated across onto the AICC2012 chronology (Bazin et al., 2013; Veres et al., 2013) using the alignment strategy of Govin et al. (2012) and the conversion tables of Lisiecki and Raymo (2005) and Parrenin et al. (2013b).

Chronological uncertainties for the MIS 5e ages of samples in this study (Table 2) vary between 2.5 and 2.7 ka. The AICC2012 chronology has an uncertainty of ±1.5 ka during MIS 5e, with an additional uncertainty of ±1 ka arising from the translation between chronologies (Capron et al., 2014). Each core sample comprises a 0.5 cm thick slice of sediment, and therefore additional age uncertainty due to integrating over the corresponding time interval in each core needs to be taken into account (see Table 2).

### 3.2. ANTA91-8 chronology

The chronology for core ANTA91-8 was constructed by aligning the magnetic susceptibility (MS) to the LR04 benthic foraminifera $\delta^{18}O$ stack (Lisiecki and Raymo, 2005) using the AnalySeries software (Paillard et al., 1996). Increased supply of terrigenous glacigenic detritus from the Antarctic continent to its margin and increased dust input from Patagonia and Australia to the pelagic SO during glacial periods resulted in higher MS values during glacial periods than interglacial periods (Bareille et al., 1994; Pugh et al., 2009; Walter et al., 2000). Tie points were selected in the MS record at the boundaries of MIS stages and sub-stages (Figure 2 & Table 3). Ages for the MIS 5 sub-stage boundaries are from Govin et al. (2009), and the ages are translated from the LR04 chronology onto the AICC2012 chronology.

The chronology for core ANTA91-8 presented in this study differs from chronologies previously
published by Ceccaroni et al. (1998) and Brambati et al. (2002), who – on the basis of [230]Thorium
measurements, subsequently adjusted by matching maxima in palaeo-productivity proxies to peak
interglacials – placed MIS 5e ~50 cm higher than in our age model (Supplementary Figure 1). Our new
chronology assigns the MS minimum from 2.65-3.05 metres below seafloor (mbsf), which comprises
a peak in organic carbon content (Ceccaroni et al., 1998), to MIS 5e. In contrast, both the Ceccaroni et
al. (1998) and Brambati et al. (2002) age models placed this MS minimum within MIS 6 (Supplementary
Figure 1), resulting in inexplicably high accumulation rates of productivity proxies during this glacial
period (Ceccaroni et al., 1998). Our new chronology is corroborated by *R. leventerae*, which occurs in
abundances <1 % in ANTA91-8 samples between 2.72 and 3.14 mbsf (Chadwick and Allen, 2021a). If
the sediments in this depth interval were deposited during MIS 6, as suggested by the Ceccaroni et al.
(1998) and Brambati et al. (2002) age models, then the *R. leventerae* abundances in the corresponding
samples should be >1 % (Zielinski et al., 2002).

| ANTA91-8 depth (mbsf) | LR04 Age (ka) | MIS stage/sub-stage boundary |
|---|---|---|
| 0.65 | 14 | 1-2 |
| 2.09 | 71 | 4-5a |
| 2.39 | 83 | 5a-b |
| 2.55 | 105 | 5c-d |
| 2.65 | 116 | 5d-e |
| 3.05 | 131.5 | 5e-6 |
| 3.17 | 136 | - |

**Table 3:** Tiepoints for ANTA91-8 chronology. The MS record for ANTA91-8 is aligned to the LR04 benthic stack using the AnalySeries software (Paillard et al., 1996).

**Figure 2:** Alignment between the MS from core ANTA91-8 (red) and the LR04 benthic $\delta^{18}O$ stack (black) using the AnalySeries software (Paillard et al., 1996). Blue squares and connecting lines mark the tiepoints between records. ⟶

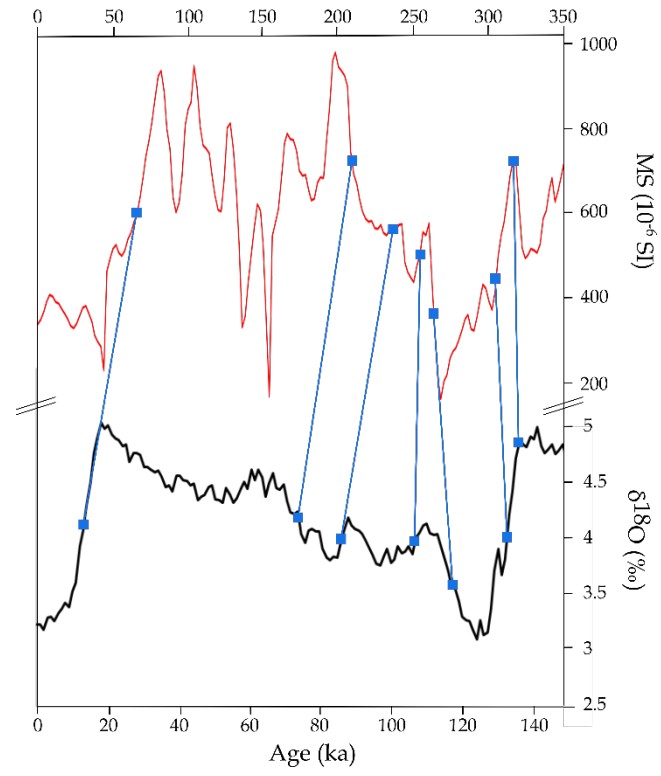

**4. Results**
The September SIC values, reconstructed using the MAT, and the FCC relative abundances are
presented for the 132-120 ka interval in all nine sediment cores (Figure 3). This interval is chosen to
capture the sea-ice signature from both the end of glacial Termination II and during 'peak' MIS 5e.

SSST data, also reconstructed using MAT, is presented over the same time interval alongside the relative abundance of *A. tabularis*.

### 4.1. Sea ice

The three Atlantic sector cores (TPC290, TPC288 and TPC287) display a N-S increasing trend in mean FCC relative abundances (2.1 ± 0.7 %, 3.1 ± 2.2 % and 4.7 ± 3.6 %) and Sept. SICs (19 ± 17 %, 25 ± 18 % and 33 ± 20 %). All three cores have low FCC relative abundances (1.2 ± 0.5 %) and Sept. SICs (8.8 ± 4.6 %) during the 131-130 ka interval, with cores TPC288 and TPC287 both reaching their minimum Sept. SIC and FCC values at this time (Figure 3). Following this interval of low Sept. SIC and FCC values, all three cores show an increase to their maximum Sept. SICs (58 ± 5 %) and FCC relative abundances (9 ± 5 %) at 127-126 ka (Figure 3). After 126 ± 2.6 ka core TPC290 displays a gradual decline in both FCC relative abundance and Sept. SIC to minimum values at 121-120 ka (Figure 3). In contrast, core TPC287 maintains high Sept. SICs (51 ± 3 %, multiple samples) throughout the 126-120 ka period as well as high (6.2 ± 1.8 %, multiple samples) FCC relative abundances, although they are lower than the 126 ± 2.6 ka peak of ~15 % (single sample) (Figure 3). Core TPC288 maintains, relative to the ~130 ± 2.5 ka minimum and ~126 ± 2.5 ka maximum, intermediate FCC (2.9 ± 0.6 %, multiple samples) and Sept. SIC (22 ± 15 %, multiple samples) values throughout the 126-120 ka interval, but the Sept. SICs are much more variable than in TPC287 (Figure 3).

All three Atlantic sector cores (TPC290, TPC288 and TPC287) have a strong match between the FCC and Sept. SIC variations ($p = 0.05$, $p < 0.01$ and $p < 0.01$ respectively), with the notable exception of the TPC287 sample at ~129 ± 2.6 ka, which has a very high Sept. SIC (86 %) but a relatively low FCC relative abundance (3.4 %). For this sample, only a single modern analog could be identified, indicating that the fossil diatom assemblage is different from almost everything in the modern reference database. The single selected analog is not chosen by the transfer function for any of the other MIS 5e samples from core TPC287, indicating that it is unlikely to be a truly representative modern analog for the MIS 5e condition at this core site. The location of this single selected analog, which is further south than any of the analogs chosen for the other MIS 5e samples from core TPC287, suggests that the fossil assemblage has been biased towards colder, heavier sea-ice conditions, probably due to dissolution or transport of the preserved assemblage. Thus, the reconstructed Sept. SIC for this sample is disregarded from the analysis. There are two MIS 5e samples in TPC290 (at 124.7 ± 2.6 ka and 122.8 ± 2.6 ka), for which none of the reference surface sediment samples were below the dissimilarity threshold (see section 2.3 for details) and thus no MAT estimate of Sept. SIC (or SSST) is given for those samples.

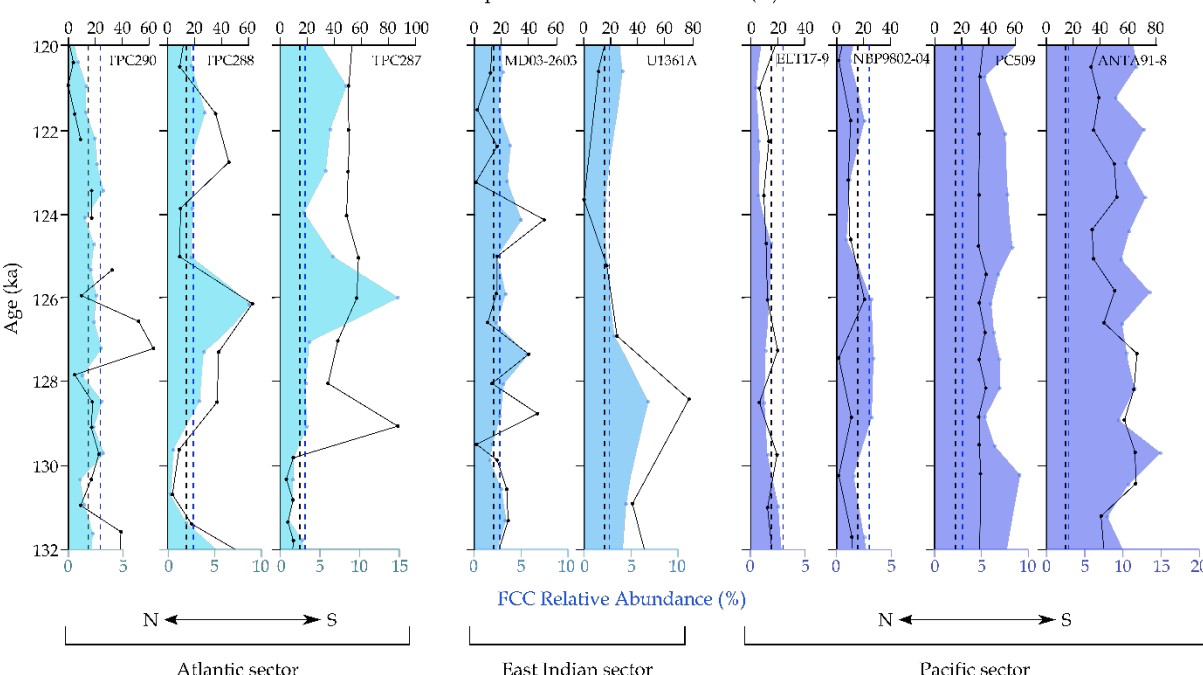

**Figure 3:** Down-core September SICs, determined using the MAT, and FCC relative abundances for the 132-120 ka interval in nine marine sediment cores. The blue shading indicates the FCC relative abundance, with the colour saturation varying between SO sectors. The solid black lines indicate the September SICs with the gaps in the TPC290 record caused by two samples being too dissimilar from all modern reference samples, so that the latter cannot be considered as analogs. Dashed lines mark the mean WSIE thresholds of 3 % FCC abundance (blue lines) and 15 % Sept. SIC (black lines). Within each SO sector cores are arranged from north to south.

To check for other potentially anomalous palaeo-reconstructions, the number of times each modern reference sample was selected as an analog were considered (Supplementary Figure 2). Fossil samples were separated into three MIS 5e-Termination II time intervals (following the approach of Chadwick et al. (2022)) and modern reference samples that are only selected as analogs for a small number (<5) of fossil samples were identified (Supplementary Figure 2). None of these less-selected reference samples are the primary or sole analog for an MIS 5e fossil sample and are therefore unlikely to result in an unrepresentative Sept. SIC (or SSST) reconstruction.

The two East Indian sector cores (MD03-2603 and U1361A) have similar average MIS 5e FCC relative abundances (3.2 ± 1 % and 3.9 ± 1.5 %) to each other but the average Sept. SIC (19 ± 15 % and 27 ± 25 %) is nearly 10 % higher in U1361A. However, the MIS 5e variability in Sept. SIC within each core is greater than this difference between the two cores. Core MD03-2603 has three Sept. SIC maxima of >40 % (single samples) during MIS 5e, at 124.1 ± 2.6 ka, 127.3 ± 2.6 ka and 128.8 ± 2.6 ka, as well as three minima of <5 % (single samples) at 121.5 ± 2.6 ka, 123.3 ± 2.6 ka and 129.5 ± 2.6 ka (Figure 3). Contrastingly, the nearby core from Hole U1361A (Figure 1) has a maximum in MIS 5e Sept. SIC (76.4 %, single sample) at 128.4 ± 2.7 ka and a minimum (0 %, single sample) at 123.7 ± 2.7 ka (Figure 3).

Together these two records suggest that the greatest MIS 5e Sept. SICs in the East Indian sector
occurred during the 129-127 ka interval and the minimum was at 123.5-121 ka (Figure 3).
Unlike the Atlantic and East Indian sectors, the four cores from the Pacific sector (ELT17-9, NBP9802-
04, PC509 and ANTA91-8) have low variability in their FCC relative abundances (1.4 ± 0.6 %, 2.3 ± 1 %,
5.8 ± 0.9 % and 11 ± 1.9 %) and Sept. SICs (13 ± 4 %, 8.4 ± 5.7 %, 34 ± 2 % and 48 ± 11 %) throughout
MIS 5e, with no pronounced maxima or minima (Figure 3). The northernmost Pacific sector core
ELT17-9 has the lowest average MIS 5e FCC relative abundance (1.4 ± 0.6 %) but the more southerly
core NBP9802-04 has the lowest average MIS 5e Sept. SIC (8.4 ± 5.7 %). The two most southerly Pacific
sector cores (PC509 and ANTA91-8) have the highest average MIS 5e Sept. SICs and FCC relative
abundances of all the cores analysed for this study.
*4.2. Sea-surface temperatures*
For the Atlantic sector cores the average MIS 5e SSSTs (3.2 ± 1.9 $^{\circ}$C, 2.7 ± 1.6 $^{\circ}$C and 2.2 ± 1.5 $^{\circ}$C) show
an inverse trend to Sept. SICs with higher values in more northerly cores. Both TPC288 and TPC287
have their highest MIS 5e SSSTs during the 131-129 ka interval (5 $^{\circ}$C and 4.3 $^{\circ}$C, respectively, multiple
samples) followed by a SSST minimum at ~126 ± 2.6 ka (0.1 $^{\circ}$C and 0.6 $^{\circ}$C, respectively, single samples)
(Figure 4). In contrast, the warmest MIS 5e SSSTs for TPC290 occur in the youngest part of the record,
with an average of 6 $^{\circ}$C in the 122-120 ka period (Figure 4). The relative abundance of *A. tabularis* in
core TPC290 shows a good consistency (p = 0.01, $R^2$ = 0.34) with the SSST pattern during MIS 5e, with
the highest relative abundances (1.3 ± 0.8 %, multiple samples) observed after 126 ± 2.6 ka (Figure 4).
The southernmost core TPC287 from the Atlantic sector shows a very poor match between MIS 5e
SSSTs and *A. tabularis* relative abundances (p = 0.3, $R^2$ = 0.09). This lack of correlation is likely due to
the scarcity of *A. tabularis* at this site throughout MIS 5e, as can be seen in modern surface sediments
(Chadwick, 2020), and thus a single valve can create a relative abundance peak that may be largely
unrelated to the SSST trends.
The East Indian sector cores have similar average SSSTs (2.8 ± 1.1 $^{\circ}$C and 2.4 ± 1.7 $^{\circ}$C). However, unlike
for the Sept. SICs (Figure 3), the MIS 5e SSST minima and maxima in cores MD03-2603 and U1361A
occur at different times (Figure 4). SSSTs in core U1361A fall to a minimum of 0.7 $^{\circ}$C (single sample) at
~128 ± 2.7 ka before rising to a maximum of 5.9 $^{\circ}$C (single sample) at ~124 ± 2.7 ka. In contrast, SSSTs
in core MD03-2603 reach an early peak of 5.9 $^{\circ}$C (single sample) at ~129.5 ± 2.6 ka and have minima
of ~1 $^{\circ}$C (single samples) at 124.1 ± 2.6 ka, 127.3 ± 2.6 ka and 128.8 ± 2.6 ka (Figure 4). Both MD03-
2603 and U1361A show a strong coherence between the MIS 5e SSSTs and the *A. tabularis* abundance
(p = 0.01, $R^2$ = 0.4 and p <0.01, $R^2$ = 0.92 respectively).

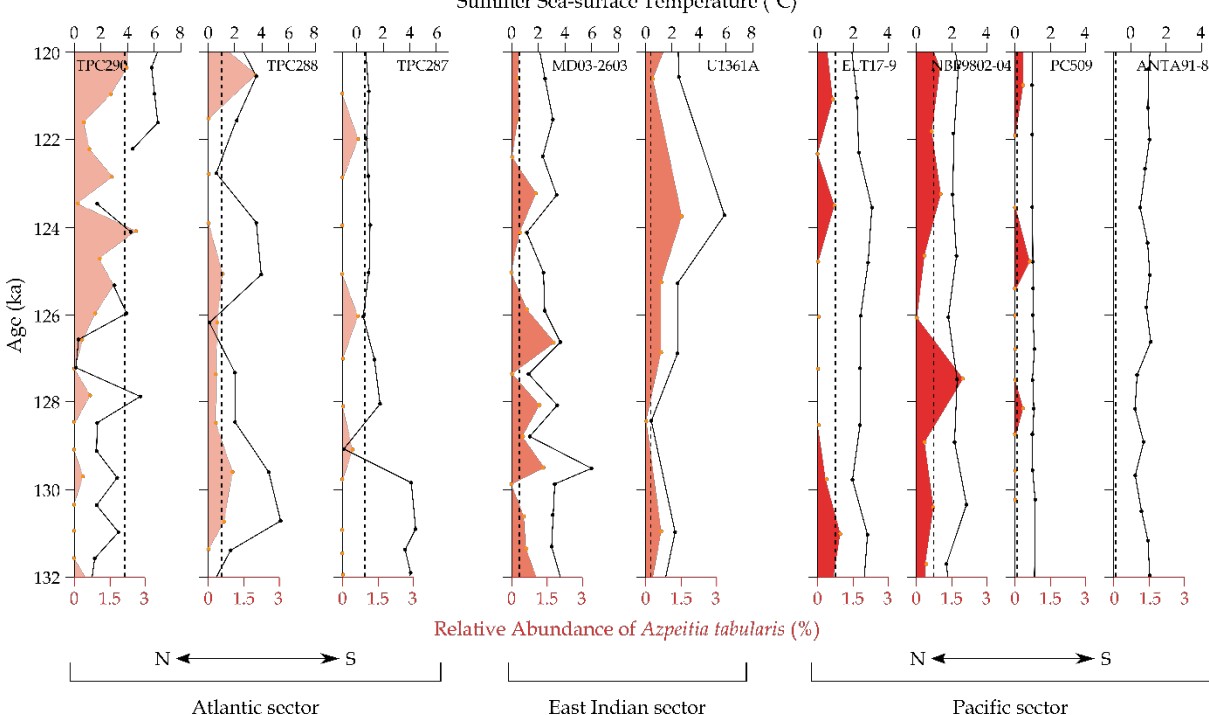

**Figure 4:** Down-core summer (January to March) SSTs, determined using the MAT, and the relative abundance of *Azpeitia tabularis* for the 132-120 ka interval in nine marine sediment cores. The red shading indicates the relative abundance of *A. tabularis*, with the colour saturation varying between SO sectors. The solid black lines indicate the SSSTs with the gaps in the TPC290 record caused by two samples being too dissimilar from all modern reference samples, so that the latter cannot be considered as analogs. Black dashed lines mark the modern (Jan-Mar, 1980-2019) SSSTs at each core site (Hersbach et al., 2019). Within each SO sector cores are arranged from north to south.

In the Pacific sector cores, SSSTs are largely consistent throughout MIS 5e, with averages of 2.5 ± 0.3 %, 2.2 ± 0.3 %, 1.03 ± 0.03 % and 0.8 ± 0.3 % (Figure 4). Although there is very little variation in MIS 5e SSSTs in all four records, both core NBP9802-04 and core PC509 reveal maximum SSSTs (2.8 °C and 1.1 °C, respectively, single samples) at ~130 ± 2.7 ka (Figure 4). None of the Pacific sector cores show a strong match between MIS 5e SSSTs and the relative abundance of *A. tabularis*. For the more southerly core PC509 this poor correlation (p = 0.65, $R^2$ = 0.02) is likely caused by the same scarcity of *A. tabularis* as for core TPC287 in the Atlantic sector.

## 5. Discussion

Both the Sept. SICs and FCC relative abundances indicate substantial differences in the pattern of MIS 5e WSIE change between the three SO sectors, most notably between the Atlantic and Pacific sectors. In all three Atlantic sector records, the FCC relative abundances and Sept. SICs indicate year-round open marine conditions and thus a poleward contraction of the mean WSIE (FCC <3 % (Gersonde and Zielinski, 2000) and Sept. SIC <15 % (Zwally et al., 2002)) during the 131-130 ka interval. This minimum is succeeded by a re-expansion of sea ice to a maximum extent in the 127-126 ka interval when all

three core sites were covered by WSI. An early minimum in MIS 5e WSIE succeeded by a maximum ~4
ka later is a consistent, but offset, pattern as the FCC relative abundance in nearby core PS2305-6
(Figure 1; 58.72 ºS, 33.04 ºW) (Bianchi and Gersonde, 2002; Chadwick et al., 2020).
We cannot rule out that the apparent retreat in Atlantic sector sea ice to a minimum during
Termination II followed by a sea-ice expansion coincident with peak Antarctic air temperatures is an
artefact caused by chronological uncertainties, with the WSIE minimum actually occurring alongside
the peak Antarctic air temperatures at ~128 ± 1.5 ka (Holloway et al., 2017; Parrenin et al., 2013a).
However, a genuine early (i.e., before 130 ka) retreat in Atlantic sector sea ice would also be consistent
with most of the Termination II and MIS 5e records from this sector analysed by Bianchi and Gersonde
(2002). Model experiments by Menviel et al. (2010) have demonstrated that during early MIS 5e the
release of vast quantities of glacial meltwater into the surface waters of the Antarctic Zone (i.e., the
region south of the Antarctic Polar Front) caused by Antarctic ice sheet deglaciation, especially the
potential partial or total loss of the West Antarctic Ice Sheet (WAIS), would have led to SST reduction
and equatorward sea-ice expansion. Importantly, this meltwater injection into the SO, which is
supported by the observation of meltwater "spikes" characterizing planktic foraminifera $\delta^{18}O$ data in
cores from the Weddell Sea continental margin during glacial-interglacial transitions (Grobe et al.,
1990), would also have resulted in a warming of subsurface waters that, in turn, would have triggered
further ocean-forced melting of the ice-sheet grounding zones, especially of the predominantly
marine-based WAIS, thus kick starting a positive feedback loop (Bronselaer et al., 2018; Menviel et al.,
2010). Because of their location within "Iceberg Alley", a main pathway of Antarctic icebergs travelling
with the clockwise Weddell Gyre from the southern Weddell Sea Embayment into the Scotia Sea
(Weber et al., 2014), core TPC290 and especially cores TPC287 and TPC288 can be expected to be
particularly sensitive for recording such meltwater supply.
In fact, the MIS 5e WSIE maximum in the Atlantic sector records coincides, within chronological
uncertainty, with higher global sea level (Kopp et al., 2013) and evidence for increased meltwater flux
in the Weddell Sea (Chadwick et al., 2022), which both indicate substantial mass loss from the
Antarctic ice sheets, consistent with findings of major ice loss in the Weddell Sea sector during MIS 5e
(Turney et al., 2020). Higher glacial meltwater fluxes associated with increased ice-sheet loss could
therefore be a major driver of the WSIE expansion in the Atlantic sector records as less saline surface
waters freeze more easily (Bintanja et al., 2013; Merino et al., 2018). The peak in FCC abundance in
core TPC287 at 126 ± 2.6 ka is primarily a peak in the abundance of *F. cylindrus* (Chadwick and Allen,
2021f). *Fragilariopsis cylindrus* generally dominates water column diatom assemblages in both ice-
covered (Burckle et al., 1987) and marginal sea-ice zones (Kang and Fryxell, 1992, 1993; Kang et al.,
1993). The occurrence of high modern *F. cylindrus* abundances in marginal sea-ice zones indicates that

this species is not purely associated with sea-ice, from which it might have been seeded when retreating, but also strongly affiliated with sea-ice melt and strong surface stratification (Cremer et al., 2003; Kang and Fryxell, 1993; von Quillfeldt, 2004). The peak in *F. cylindrus* abundances at 126 ± 2.6 ka in core TPC287, separate from any notable increase in *F. curta* abundance, therefore supports an increased glacial meltwater signal at this time.

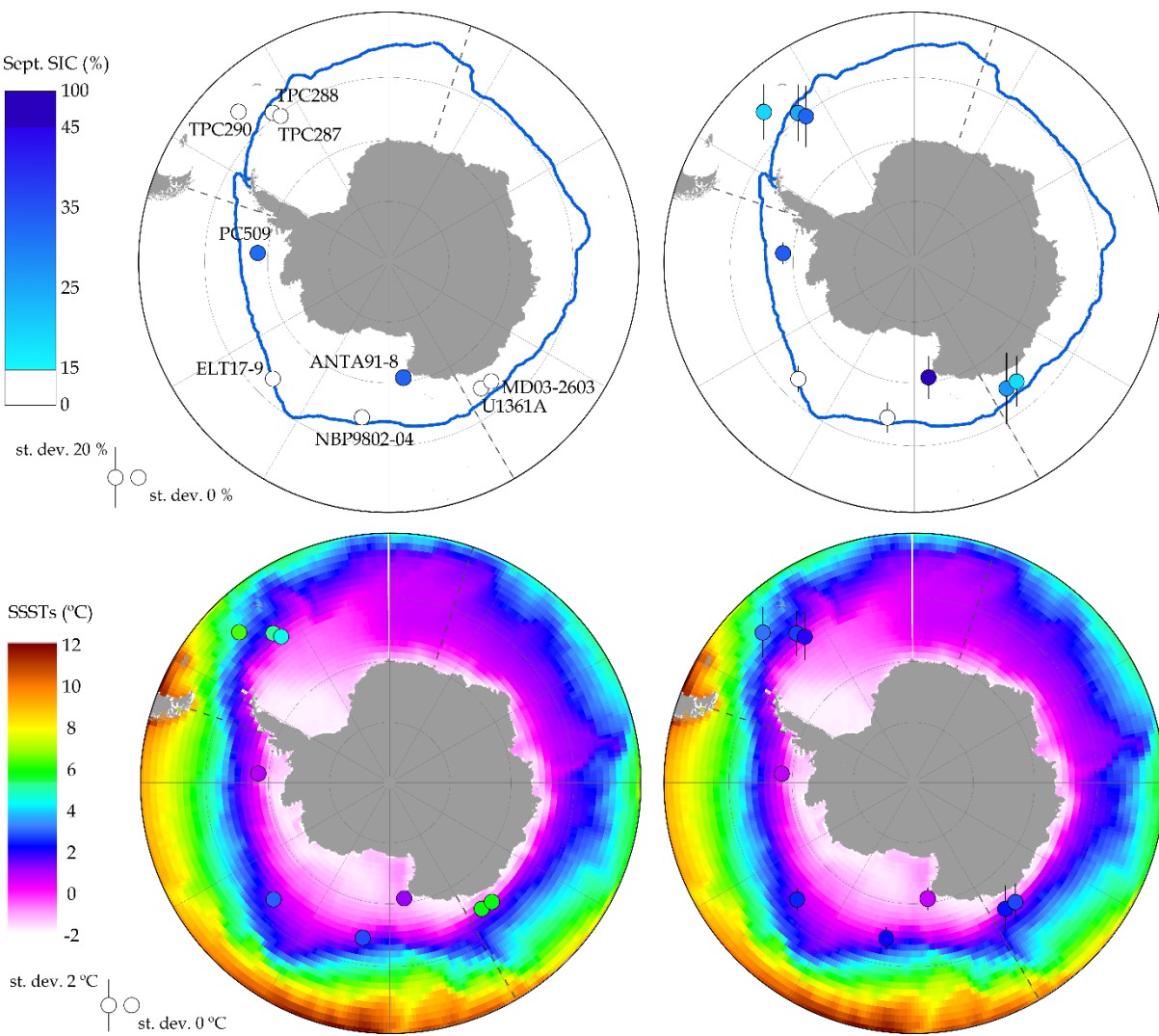

**Figure 5:** Maps of MIS 5e SSSTs and Sept. SICs for the nine core sites compared with the modern conditions. On all maps the SO sector boundaries are marked with dashed lines. **Top left:** Minimum MIS 5e Sept. SIC for each core site (coloured circles) compared to the modern (1981-2010) 15 % September sea-ice extent (blue line) (Fetterer et al., 2017). **Top right:** Average MIS 5e Sept. SICs (coloured circles) and standard deviations (vertical bars) at each core site compared to the modern (1981-2010) 15 % September sea-ice extent (blue line) (Fetterer et al., 2017). **Bottom left:** Maximum MIS 5e SSSTs for each core site (coloured circles) compared to modern (Jan-Mar, 1980-2019) SSSTs (Hersbach et al., 2019). **Bottom right:** Average MIS 5e SSSTs (coloured circles) and standard deviations (vertical bars) for each core site compared to modern (Jan-Mar, 1980-2019) SSSTs (Hersbach et al., 2019). Core data are given in Supplementary Table 1.

The discrepancy between Sept. SICs and FCC relative abundances at ~127 ± 2.6 ka in core TPC290 (Figure 3) is likely due to increased *Chaetoceros* resting spore (rs.) abundance at this time (Chadwick and Allen, 2021h). This *Chaetoceros* rs. abundance increase is also observed in the nearby core PS2305-6 (Bianchi and Gersonde, 2002) and is inferred to be caused by higher meltwater and iceberg flux at this time (Bianchi and Gersonde, 2002; Crosta et al., 1997). For core TPC290, there is a scarcity of modern analogs from the Scotia Sea region (Figure 1) and thus, the high *Chaetoceros* rs. abundances in MIS 5e samples are associated with modern analogs from sites along the Antarctic Peninsula, where SICs are greater than in the Scotia Sea.

Atlantic sector SSSTs reach their maxima during Termination II before a substantial drop coincident with the peak Antarctic air temperatures in ice cores (Parrenin et al., 2013a). As with the down-core Sept. SIC profiles, this offset may result from chronological uncertainties, with the highest SSSTs actually occurring alongside peak Antarctic air temperatures at ~128 ± 1.5 ka. However, air temperature and SST reconstructions from the Antarctic Peninsula and Scotia Sea have shown that during Termination I temperatures peaked at higher values than during the Holocene (Mulvaney et al., 2012; Xiao et al., 2016), thus, our records could indicate an equivalent early warming during Termination II for this region. Also, if the high air temperatures at ~128 ± 1.5 ka caused substantial Antarctic ice sheet loss, then the cold SSSTs in our ice-sheet proximal records at this time could, as discussed above, actually reflect major input of cold and fresh meltwater not recorded in cores further north.

In the East Indian sector, core MD03-2603 has an average MIS 5e Sept. SIC (25 ± 18 %) and FCC relative abundance (3.2 ± 1 %) indicative of a location just south of the mean WSIE (Figures 3 & 5) but with multiple maxima and minima contributing to the high variability. MIS 5e Sept. SICs and FCC relative abundances in the nearby core U1361A indicate that it was located within the seasonal sea-ice zone from 132-126 ka before the mean WSIE retreated to the south of this location (Figure 3 & Supplementary Figure 3). The different patterns in MIS 5e Sept. SIC and SSSTs between cores MD03-2603 and U1361A are likely due to the different age resolution of the samples, with two of the Sept. SIC maxima in MD03-2603 occurring in the 129-127 ka interval coincident with the U1361A Sept. SIC maximum, and likewise, two of the Sept. SIC minima in MD03-2603 occurring in the 124-121 ka period concurrent with the minimum Sept. SIC in core U1361A (Figure 3). The different age resolution of samples in MD03-2603 and U1361A is primarily due to the lower sedimentation rate (Table 2) at site U1361A, and thus a sample from this core spans more time than in core MD03-2603.

None of the Pacific sector cores show pronounced minima or maxima in their MIS 5e FCC and Sept. SIC records (Figure 3), indicating a less variable WSIE in this sector compared to the Atlantic and Indian

sectors (Figure 3). The Pacific sector cores PC509 and ANTA91-8 are also the only cores in this study
which are covered by WSI for the entirety of MIS 5e (Figure 3 & 5). The position of these cores south
of the mean WSIE throughout MIS 5e is significant as they are the first published marine records from
within the seasonal sea-ice zone and able to constrain the poleward limit of the MIS 5e minimum WSIE
(Chadwick et al., 2020). Cores ELT17-9 and NBP9802-04 are the only records in this study with average
MIS 5e Sept. SICs <15 % (Figure 5), indicating they were located north of the mean WSIE for the
majority of the 132-120 ka period, with core ELT17-9 having been located closer to the MIS 5e mean
WSIE. The FCC relative abundances for cores ELT17-9 and NBP9802-04 also indicate that both sites
were predominantly positioned north of the mean WSIE during MIS 5e (Figure 3) but suggest that core
NBP9802-04 was located closer to the MIS 5e mean WSIE.
The reconstructed MIS 5e Sept. SICs for site ELT17-9 are higher than for site NBP9802-04 (Figure 3)
which is likely related to the higher abundance of *Chaetoceros* rs. in core ELT17-9 when compared to
core NBP9802-04 (Chadwick and Allen, 2021b, d). The *Chaetoceros* rs. group is associated with both
WSI (Armand et al., 2005) and meltwater stratification (Crosta et al., 1997), and high abundances of
*Chaetoceros* rs. in Ross Sea sediments deposited during past interglacial periods have been linked to
increased upwelling and subsequent meltwater stratification within the Ross Sea Gyre (Kim et al.,
2020). The high *Chaetoceros* rs. abundance in core ELT17-9 during MIS 5e could therefore indicate an
north-eastward shift of the Ross Sea Gyre from its modern day position (Dotto et al., 2018) and an
accompanying displacement of meltwater circulation (Merino et al., 2016) and the WSI edge in the
Pacific sector. It is also possible that the reduced Pacific sector WSIE during MIS 5e is associated with
earlier seasonal sea-ice retreat during the austral spring and a longer open-ocean season, promoting
a stronger spring bloom signal, of which the *Chaetoceros* group is a major component (Leventer,

399    1991).

The average MIS 5e SSSTs in the nine cores are ~1-2 °C warmer than the modern SSSTs (Figure 5),
consistent with the SST anomalies presented in Chadwick et al. (2020) and Capron et al. (2014).
However, the SSST records in the Atlantic and East Indian sectors have large variability with maximum
SSSTs that are 2-4 °C higher than the MIS 5e average SSSTs (Figure 5). Maximum MIS 5e SSSTs in the
Atlantic and East Indian sectors were therefore ~3-5 °C warmer than modern SSSTs (Figure 5), which
is a much larger SSST anomaly than in the Antarctic Zone records presented in Chadwick et al. (2020),
and marks a ~5 degrees latitude poleward shift in SSST isotherms relative to the present. Unlike the
Atlantic and East Indian sectors, the Pacific sector core records indicate low variability in MIS 5e SSSTs
with peak values 0-2 °C warmer than present (Figure 5) marking a poleward shift in SSST isotherms of
<3° latitude.
Within their chronological uncertainties (Table 2), cores TPC288, TPC287, MD03-2603, ELT17-9,
NBP9802-04 and PC509 all reach minimum MIS 5e Sept. SICs synchronously (Supplementary Table 1)
and coincident with the peak in Antarctic air temperatures and minimum in EPICA Dome C (EDC) sea-
salt sodium flux ($Na_{ss}$) at ~128 ± 1.5 ka (Holloway et al., 2017; Wolff et al., 2006). The two East Indian
sector core records reach a minimum MIS 5e WSIE (and maximum SSST in core U1361A) ~4.5 ka after
the $Na_{ss}$ minimum in Antarctic ice cores, outside of the combined chronological uncertainties of the
sediment cores (Table 2) and AICC2012 ice core chronology (Bazin et al., 2013). Although the duration
of the SSST maximum, and accompanying WSIE minimum, in core MD03-2603 is short, it occurs within
chronological error of the maximum air temperatures in Antarctic ice cores (Figures 3 & 4).
Satellite era trends in Antarctic winter SIC (Hobbs et al., 2016) are largely consistent with the patterns
observed during MIS 5e. Northern Weddell Sea winter SIC has declined by 5-10 % per decade in the
satellite era (Hobbs et al., 2016) indicating a sensitivity to warming consistent with the early retreat
of MIS 5e sea ice in this region. Similarly, winter SICs in the Pacific sector have remained stable, or
even slightly increased, during the satellite era (Hobbs et al., 2016) which is in agreement with the
stability of the Pacific sector WSIE throughout MIS 5e. In recent decades, Bellingshausen Sea summer
sea ice has decreased, whilst WSIE has stayed stable (Hobbs et al., 2016; Parkinson, 2019). The MIS 5e
Sept. SICs and SSSTs (as a proxy for summer sea ice) imply that the MIS 5e WSIE in the Bellingshausen
Sea is similar to the modern but the summer sea-ice extent was reduced. The northern part of the
Ross Sea is a region in which the modern and MIS 5e trends differ, with recent winter SIC increases of
10-15 % per decade contrasting with the MIS 5e WSIE reduction observed at site NBP9802-04.

### 430    6. **Wider implications**

During MIS 5e the three SO sectors display heterogeneous responses in WSIE and SSSTs, which may
guide our predictions of the impact of future warming on the Antarctic region. The prominent early
(131-130 ka) minimum in WSIE and coinciding maximum in SSSTs for the two southerly Atlantic sector
cores (TPC288 and TPC287, Figure 3) is associated with a mean WSI edge located at least 3-5 ° south
of its modern position. This substantial reduction in WSIE and seasonal sea-ice cover would have
reduced brine rejection and likely decreased the rates of deep and bottom water formation in the
Weddell Sea, causing a warming of the abyssal waters (Bouttes et al., 2010; Marzocchi and Jansen,
2019). Deep water warming would have promoted the basal melting and retreat of Weddell Sea ice
shelves and marine terminating ice streams and caused substantial Antarctic ice sheet mass loss
(Hellmer et al., 2012; Rignot et al., 2019; Wahlin et al., 2021). We hypothesise that substantial mass
loss from the Weddell Sea sector of the WAIS (Turney et al., 2020) drove the Atlantic sector WSI
resurgence at ~126 ± 2.6 ka, as suggested by the model experiments of Menviel et al. (2010), and
contributed to the global sea-level rise at this time (Kopp et al., 2013; Sime et al., 2019).
Variations in the WSIE and SSST records between the East Indian sector cores MD03-2603 and U1361A
are due to the differences in sampling resolution, with the MD03-2603 record indicating multiple
relatively short duration WSIE and SSST oscillations during MIS 5e. The U1361A record seems to
present an averaged signal of these oscillations with a greater frequency of warm periods with
reduced WSIE after 125 ± 2.7 ka. Along the modern Wilkes Land margin the Antarctic Circumpolar
Current (ACC) flows much closer to the continent than in other regions (Tamsitt et al., 2017) and the
MIS 5e record in core MD03-2603 could therefore suggest multiple intervals when the ACC was
displaced to the south of its modern position. A more southerly ACC in this region would have caused
a poleward shift in precipitation fields and resulted in drier conditions across Southern Australia (Liu
and Curry, 2010; Saunders et al., 2012), a trend that can already be observed under a modern warming
climate (CSIRO, 2018). A southerly shift of the ACC would also increase the advection of warmer
Circumpolar Deep Water onto the Antarctic continental shelf (Fogwill et al., 2014), promoting periods
of high basal melting and ice sheet retreat in Wilkes Land during MIS 5e, as supported by Wilson et al.
457 (2018).

In contrast to the Atlantic and East Indian sectors, the Pacific sector records indicate a more stable
WSIE throughout MIS 5e. The MIS 5e Sept. SIC records of cores ELT17-9 and NBP9802-04 indicate a
poleward shift in the mean WSI edge by at least 2 ° of latitude relative to the modern. The PC509
record indicates a southerly shift in the mean WSI edge by <2 ° latitude. This highlights a seemingly
greater resilience of sea ice in the Bellingshausen Sea, with the WSI edge remaining north of 68 °S
throughout MIS 5e, possibly in response to major glacial meltwater release from the Bellingshausen
Sea drainage basin of the WAIS. In the modern Pacific sector the WSIE is strongly constrained by the
southern extent of the ACC and the configuration of the Ross Sea Gyre (Benz et al., 2016; Nghiem et
al., 2016). An uneven poleward constriction of the ACC across the Pacific sector during MIS 5e could
therefore help explain the differing WSI retreat in this sector, with greater poleward migration of the
ACC and reduction in the Ross Sea Gyre northward extent in the western Pacific sector than in the
eastern Pacific sector. However, unlike in the East Indian sector, there is no evidence for millennial-
scale migration of the ACC across the Pacific sector. The stable and persistent WSIE in the Pacific sector
during MIS 5e may have resulted from major WAIS deglaciation (Menviel et al., 2010), but then
protected further melting of ice shelves in the Ross, Amundsen and Bellingshausen seas which
buttressed ice grounded further upstream (Massom et al., 2018). This buttressing may have acted as
a stabilising factor preventing total loss of the WAIS during MIS 5e, with the majority of its deep
subglacial basins terminating in the Ross, Amundsen and Bellingshausen Seas (Gardner et al., 2018).

The sensitivity of Weddell Sea WSI to warmer climates could have substantial implications for the SO biosphere given the high rates of primary productivity in this region today (Vernet et al., 2019). Whilst a future reduction in WSIE and increase in glacial meltwater flux would be expected to promote primary productivity in the western part of the Weddell Sea (de Jong et al., 2012), the higher SSTs would not favour key trophic intermediaries, e.g. Antarctic krill (*Euphausia superba*) (Atkinson et al., 2017; Siegel and Watkins, 2016), and would therefore negatively affect megafauna at higher trophic levels (Hill et al., 2013). The impacts of warming and reduced WSIE on the SO food web are seen along the Antarctic Peninsula in the present day, with a recent shift in phytoplankton community structure from diatoms to smaller cryophytes, which are less efficiently grazed by Antarctic krill (Mendes et al., 2018; Moline et al., 2004). Future WSI edge retreat, at equivalent levels to MIS 5e, would also negatively impact upon modern sea-ice obligate species, such as Emperor and Adélie Penguins (Cimino et al., 2013; Jenouvrier et al., 2005).

## 7. **Conclusions**

Similarly to the modern SO (Parkinson, 2019), WSIE trends during MIS 5e show both spatial and temporal heterogeneity. The Atlantic and East Indian sectors display more variable WSIE and SSTs during MIS 5e than the Pacific sector. High Atlantic sector environmental variability during MIS 5e is attributed to high glacial meltwater release from the Weddell Sea drainage sector of the WAIS, whereas the high variability in the East Indian sector is attributed to large latitudinal migrations of the ACC flow bands occurring on a millennial timescale. In contrast, the stability of the Pacific sector WSIE may be due to the local bathymetric pinning of the ACC limiting the possible poleward displacement of the ACC during MIS 5e.

The greater MIS 5e WSIE reduction in the Atlantic sector compared to the Pacific sector is consistent with recent model simulations (Holloway et al., 2017). Most of the core records in this study reach their minimum WSIE at the same time, i.e., within chronological uncertainties, as the 128 ± 1.5 ka minimum in Antarctic ice core $Na_{ss}$ flux (Wolff et al., 2006), with only cores TPC290 and U1361A indicating a later WSIE minimum (Figure 3 & Supplementary Figure 3). The apparent high sensitivity of Weddell Sea WSIE, and apparent resilience of Bellingshausen Sea WSIE, to warmer than present climates is unexpected from the recent observational trends (Hobbs et al., 2016; Parkinson, 2019), but may be related to regionally variable influx of glacial meltwater and its advection around the Antarctic continent. Our study highlights the importance of reconstructing palaeoenvironmental conditions around Antarctica during past warm periods, such as MIS 5e, for understanding how the Antarctic and SO regions respond to warmer climates on longer than decadal timescales.

### Data availability

Full diatom count data for all samples are available from the NERC EDS UK Polar Data Centre (Chadwick and Allen, 2021a, b, c, d, e, f, g, h, i). Sept. SIC and SSST data for all samples, produced using the MAT transfer function, are available from PANGAEA (*in press*).

### Author contribution

**MC** – Data Curation, Investigation, Visualization, Writing – original draft preparation; **CA** – Conceptualization, Project administration, Resources, Supervision, Writing – review & editing; **LS** – Conceptualization, Supervision, Writing – review & editing; **XC** – Formal analysis, Methodology, Resources, Writing – review & editing; **CDH** – Resources, Writing – review & editing.

### Competing interests

The authors declare they have no conflict of interest.

### Acknowledgements

Funding for this work was provided by The Natural Environmental Research Council [grant number NE/L002531/1]. The British Ocean Sediment Core Research Facility (BOSCORF) is thanked for supplying sediment samples for core TPC287 and multi-sensor core logging of core PC509. We thank the Lamont-Doherty Core Repository of Lamont-Doherty Earth Observatory for providing sediment sample material for core NBP9802-04 (IGSN – DSR0003YW). The International Ocean Discovery Program (IODP) is thanked for providing the sample material for core U1361A. We also thank the Oregon State University Marine and Geology Repository for providing sediment samples for core ELT17-9, the Sorting Centre of MNA-Trieste (Italy) for providing sediment samples for core ANTA91-8 and S.J. Crowhurst from the Department of Earth Sciences, University of Cambridge (UK), for X-ray fluorescence scanning of core PC509.

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
