# Peer review of "Reconstructing Antarctic winter sea-ice extent during Marine Isotope"

_Climate of the Past, 2021_

## Author Comment (AC1)

Dear Reviewer & Editor,

We thank the reviewer for her/his supportive and constructive comments on our manuscript. Please see below, in blue, our detailed responses to the comments.

I commend the authors for the scale and complexity of this work, which reflects an incredible amount of work from start to finish. The overall aim of this project, to evaluate palaeoceanographic conditions (winter sea ice and SST) during MIS 5e, in comparison to average modern conditions, is well conceived and of broad scale interest. This time period presents an appropriate test case for comparison, in terms of understanding/ anticipating near future conditions of the Southern Ocean, and the ramifications on an array of variables, ranging from changes in bottom water formation to ecosystem scale changes. For this reason, I found the introduction and conclusion to be especially useful and aimed toward wide audience, including those with less specific interest in the details of the diatom work. The data are well-illustrated and clear, easy to follow; thanks for the common x-axis scaling.

Specific comments:

1. I realize that the authors are limited by the cores available, that are suitable for this study – core length and time scale covered, resolution, and diatom preservation. In terms of future work, identification of key missing pieces might be helpful, with a very limited ability to truly evaluate the Indian Ocean sector of the Southern Ocean, with no cores reflecting almost this entire sector, which is about half of the studied area. The two cores analyzed reflect only the edge of this sector and are relatively high latitude. Given that, it is difficult to make substantive conclusions about this sector. This is just an observation, not a criticism. However, it might be a good idea to re-frame the term "Indian Ocean sector" which really isn't well-addressed geographically.

We agree that the use of "Indian Ocean sector" was misleading given the locations of the cores. We reiterate that none of the published cores from the Indian & west Pacific sectors of the Southern Ocean (Crosta et al. 2004, Ferry et al. 2015, Nair et al. 2019, Chadwick et al. 2020, Ghadi et al. 2020, Jones et al. 2021, *in review*) are located far enough south to record sea ice during the LIG, which is the focus of this present study. As the modern sea-ice edge lies at around 62 °S in the eastern Indian sector, only cores proximal to the Antarctic continent allow to infer WSI during the LIG. Obviously, similar studies in other regions off East Antarctica must be conducted to provide a basin-wide view of sea-ice conditions in the Indian sector during the LIG. For these reasons, we will use "East Indian Ocean sector" in the revised manuscript to account for the specific location of our cores.

2. I appreciate the reluctance to overinterpret, especially when the environmental controls on some species, or species groups, is more complicated than temperature and/or sea ice. The authors allude to this for example, in noting the unusual abundance of Fragilariopsis separanda, for example, on page 9-10 (lines 193-203). This impacts their statistical analysis and interpretation, yet clearly reflects something different. Thanks to the authors for pointing this out – yet one more species to re-evaluate. And despite the very common use of F. curta + F. cylindrus as a sea ice indicator, their differing distribution in the modern ocean suggests that the story is more complex. How confident are the authors in suggesting that F. cylindrus is associated with sea ice meltwater? I suggest adding reference to several older papers, that might strengthen this association:

Kang, S.-H., Fryxell, G.A., 1992, Fragilariopsis cylindrus (Grunow) Krieger: The most

abundant diatom in water column assemblages of Antarctic marginal ice-edge zones, Polar Biology, 12, 6-7, 609-627.

Kang, S.-H., Fryxell, G.A., 1993, Phytoplankton in the Weddell Sea, Antarctica: composition, abundance and distribution in water-column assemblages of the marginal ice-edge zone during austral autumn, Marine Biology, 116, 335-352.

Kang, S-H., Fryxell, G.A., Roelke, D.L., 1993, Fragilariopsis cylindrus compared with other species of the diatom family Bacillariaceae in Antarctic marginal ice edge zones, Nova Hedwigia, 106, 335-352.

We will add these additional references along with a couple of sentences on *F. cylindrus* ecology to strengthen our argument:

*"The peak in FCC abundance in core TPC287 at 126 ± 2.6 ka is primarily a peak in the abundance of* F. cylindrus *(Chadwick & Allen 2021).* F. cylindrus *dominates modern diatom assemblages in both ice-covered and open ocean locations in the marginal sea-ice zone (Kang & Fryxell 1992, 1993, Kang et al. 1993). The occurrence of high modern* F. cylindrus *abundances in locations not covered by sea ice at any point during that season indicates that this species is not purely associated with sea-ice extent but also affiliated with sea-ice melt and strong surface stratification (Kang & Fryxell 1993, Cremer et al. 2003, von Quillfeldt 2004). The peak in* F. cylindrus *abundances at 126 ± 2.6 ka in core TPC287, separate from any notable increase in* F. curta *abundance, therefore supports an increased glacial meltwater signal at this time.*

3. Table 2: are the +/- values overly precise, especially given bioturbation? In lines 140-142, the age uncertainty is widened, given the thickness of the sample interval (which pretty narrow, only 0.5 cm).

We will amend the age uncertainties to a single decimal place, as is used throughout the rest of the manuscript.

4. How does this paper compare to the Chadwick et al. paper that is in review? Without seeing both it is difficult to evaluate the unique contributions of each.

The Chadwick et al. paper in review compares diatom abundances in MIS 5e sediments to the abundances in seafloor surface sediments and does not include any quantitative transfer function reconstructions of sea ice or SSTs. We therefore believe that the present study gives much more information on sea-ice conditions at the LIG, and the potential drivers, than the Chadwick et al. paper in review in Marine Micropaleontology.

5. Line 275 – Interesting comment regarding the abundance of Chaetoceros resting spores in TPC290, such that the analog is closer to modern day Antarctic Peninsula. Lines 302-309, another reference to higher Chaetoceros, this time in core ELT17-9. I wonder if this might be associated with earlier timing of sea ice breakout in the spring, a longer open water season with a stronger spring bloom signal? Or upwelling? Or both?

We agree that these are possible additional explanations for the higher *Chaetoceros* rs. abundances and will expand the discussion to include these alternative causes, especially with regard to the Pacific sector core ELT17-9.

6. Lines 383-391, discussion of changing ecosystem due to sea ice changes in the Weddell Sea sector: Consider references to:

Moline et al. 2004, Alteration of the food web along the Antarctic Peninsula in response to a warming trend. Global Change Biology. 10. 1973-1980 and Mendes et al., 2018, New insights on the dominance of cryptophytes in Antarctic coastal waters: A case study in Gerlache Strait. DSR II, 149, 161-170.

I realize that both these papers are from the western side of the Antarctic Peninsula but note the impact of meltwater and a warming ocean on primary producers and impacts that cascade through the food web.

We will expand the discussion on ecosystem impacts to further detail the possibly effects of warming and meltwater release on the wider food web.

7. Line 395 - remove parenthesis since this is an important consideration. In general, if it's important enough to be stated, no parentheses necessary.

We will remove the parentheses.

**Rebuttal references:**

Chadwick M. & Allen C.S. 2021. Marine Isotope Stage 5e diatom assemblages in marine sediment core TPC287 (-60.31 ºN, -36.65 ºE, Cruise JR48) *UK Polar Data Centre, Natural Environment Research Council, UK Research & Innovation*.

Chadwick M., Allen C.S., Sime L.C. & Hillenbrand C.D. 2020. Analysing the timing of peak warming and minimum winter sea-ice extent in the Southern Ocean during MIS 5e. *Quaternary Science Reviews*, **229**: 106134.

Cremer H., Roberts D., McMinn A., Gore D. & Melles M. 2003. The Holocene Diatom Flora of Marine Bays in the Windmill Islands, East Antarctica. *Botanica Marina*, **46** (1): 82-106.

Crosta X., Sturm A., Armand L. & Pichon J.-J. 2004. Late Quaternary sea ice history in the Indian sector of the Southern Ocean as recorded by diatom assemblages. *Marine Micropaleontology*, **50** (3-4): 209-223.

Ferry A.J., Crosta X., Quilty P.G., Fink D., Howard W. & Armand L.K. 2015. First records of winter sea ice concentration in the southwest Pacific sector of the Southern Ocean. *Paleoceanography*, **30** (11): 1525-1539.

Ghadi P., Nair A., Crosta X., Mohan R., Manoj M.C. & Meloth T. 2020. Antarctic sea-ice and palaeoproductivity variation over the last 156,000 years in the Indian sector of Southern Ocean. *Marine Micropaleontology*, **160**: 101894.

Jones J., Kohfeld K., Bostock H., Crosta X., Liston M., Dunbar G., Chase Z., Leventer A., Anderson H. & Jacobsen G. 2021, *in review*. Sea Ice Changes in the Southwest Pacific Sector of the Southern Ocean During the Last 140,000 Years. *Climate of the Past Discussions [preprint]*.

Kang S.-H. & Fryxell G.A. 1992. *Fragilariopsis cylindrus* (Grunow) Krieger: The most abundant diatom in water column assemblages of Antarctic marginal ice-edge zones *Polar Biology*, **12** (6-7): 609-627.

Kang S.-H. & Fryxell G.A. 1993. Phytoplankton in the Weddell Sea, Antarctica: composition, abundance and distribution in water-column assemblages of the marginal ice-edge zone during austral autumn. *Marine Biology*, **116**: 335-348.

Kang S.-H., Fryxell G.A. & Roelke D.L. 1993. *Fragilariopsis cylindrus* compared with other species of the diatom family Bacillariaceae in Antarctic marginal ice-edge zones. *Nova Hedwigia*, **106**: 335-352.

Nair A., Mohan R., Crosta X., Manoj M.C., Thamban M. & Marieu V. 2019. Southern Ocean sea ice and frontal changes during the Late Quaternary and their linkages to Asian summer monsoon. *Quaternary Science Reviews*, **213**: 93-104.

von Quillfeldt C. 2004. The diatom *Fragilariopsis cylindrus* and its potential as an indicator species for cold water rather than for sea ice. *Vie et Milieu / Life & Environment*, **54** (2-3): 137-143.

---

## Author Response (AR1)

**Author's Response**

Dear Editor and Reviewers,

We thank both reviewers for their thorough and constructive comments on our manuscript. Please see below, in blue, our detailed responses to the comments. We have also included a revised manuscript with the changes highlighted in yellow.

**Reviewer #1**

I commend the authors for the scale and complexity of this work, which reflects an incredible amount of work from start to finish. The overall aim of this project, to evaluate palaeoceanographic conditions (winter sea ice and SST) during MIS 5e, in comparison to average modern conditions, is well conceived and of broad scale interest. This time period presents an appropriate test case for comparison, in terms of understanding/anticipating near future conditions of the Southern Ocean, and the ramifications on an array of variables, ranging from changes in bottom water formation to ecosystem scale changes. For this reason, I found the introduction and conclusion to be especially useful and aimed toward wide audience, including those with less specific interest in the details of the diatom work. The data are well-illustrated and clear, easy to follow; thanks for the common x-axis scaling.

Specific comments:

1. I realize that the authors are limited by the cores available, that are suitable for this study – core length and time scale covered, resolution, and diatom preservation. In terms of future work, identification of key missing pieces might be helpful, with a very limited ability to truly evaluate the Indian Ocean sector of the Southern Ocean, with no cores reflecting almost this entire sector, which is about half of the studied area. The two cores analyzed reflect only the edge of this sector and are relatively high latitude. Given that, it is difficult to make substantive conclusions about this sector. This is just an observation, not a criticism. However, it might be a good idea to re-frame the term "Indian Ocean sector" which really isn't well-addressed geographically.

We agree that the use of "Indian Ocean sector" was misleading given the locations of the cores. We reiterate that none of the published cores from the Indian & west Pacific sectors of the Southern Ocean (Crosta et al. 2004, Ferry et al. 2015b, Nair et al. 2019, Chadwick et al. 2020, Ghadi et al. 2020, Jones et al. 2021, *in review*) are located far enough south to record sea ice during the LIG, which is the focus of this present study. As the modern sea-ice edge lies at around 62 °S in the eastern Indian sector, only cores proximal to the Antarctic continent allow us to infer WSI during the LIG. Obviously, similar studies in other regions off East Antarctica must be conducted to provide a basin-wide view of sea-ice conditions in the Indian sector during the LIG. For these reasons, we have used "East Indian Ocean sector" in the revised manuscript to account for the specific location of our cores.

2. I appreciate the reluctance to overinterpret, especially when the environmental controls on some species, or species groups, is more complicated than temperature and/or sea ice. The authors allude to this for example, in noting the unusual abundance of Fragilariopsis separanda, for example, on page 9-10 (lines 193-203). This impacts their statistical analysis and interpretation, yet clearly reflects something different. Thanks to the authors for pointing this out – yet one more species to re-evaluate. And despite the very common use of F. curta + F. cylindrus as a sea ice indicator, their differing distribution in the modern ocean suggests that the story is more complex. How confident are the authors in suggesting that F. cylindrus is associated with sea ice meltwater? I suggest adding reference to several older papers, that might strengthen this association:

Kang, S.-H., Fryxell, G.A., 1992, Fragilariopsis cylindrus (Grunow) Krieger: The most abundant diatom in water column assemblages of Antarctic marginal ice-edge zones, Polar Biology, 12, 6-7, 609-627.

Kang, S.-H., Fryxell, G.A., 1993, Phytoplankton in the Weddell Sea, Antarctica: composition, abundance and distribution in water-column assemblages of the marginal ice-edge zone during austral autumn, Marine Biology, 116, 335-352.

Kang, S-H., Fryxell, G.A., Roelke, D.L., 1993, Fragilariopsis cylindrus compared with other species of the diatom family Bacillariaceae in Antarctic marginal ice edge zones, Nova Hedwigia, 106, 335-352.

*We have added these additional references along with a couple of sentences on F. cylindrus ecology to strengthen our argument:*

*"Fragilariopsis cylindrus generally dominates water column diatom assemblages in both ice-covered (Burckle et al., 1987) and marginal sea-ice zones (Kang and Fryxell, 1992, 1993; Kang et al., 1993). The occurrence of high modern F. cylindrus abundances in marginal sea-ice zones indicates that this species is not purely associated with sea-ice, from which it might have been seeded when retreating, but also strongly affiliated with sea-ice melt and strong surface stratification (Cremer et al., 2003; Kang and Fryxell, 1993; von Quillfeldt, 2004). The peak in F. cylindrus abundances at 126 ± 2.6 ka in core TPC287, separate from any notable increase in F. curta abundance, therefore supports an increased glacial meltwater signal at this time."*

3. Table 2: are the +/- values overly precise, especially given bioturbation? In lines 140-142, the age uncertainty is widened, given the thickness of the sample interval (which pretty narrow, only 0.5 cm).

*We have amended the age uncertainties to a single decimal place, as is used throughout the rest of the manuscript.*

4. How does this paper compare to the Chadwick et al. paper that is in review? Without seeing both it is difficult to evaluate the unique contributions of each.

*The Chadwick et al. paper in review (meanwhile accepted and published, see reference list at the end of this response) compares diatom abundances in MIS 5e sediments to the abundances in seafloor surface sediments and does not include any quantitative transfer function reconstructions of sea ice or SSTs. We therefore believe that the present study gives much more information on sea-ice conditions at the LIG, and the potential drivers, than Chadwick et al. (2022).*

5. Line 275 – Interesting comment regarding the abundance of Chaetoceros resting spores in TPC290, such that the analog is closer to modern day Antarctic Peninsula. Lines 302-309, another reference to higher Chaetoceros, this time in core ELT17-9. I wonder if this might be associated with earlier timing of sea ice breakout in the spring, a longer open water season with a stronger spring bloom signal? Or upwelling? Or both?

*We agree that these are possible additional explanations for the higher Chaetoceros rs. abundances and have expanded the discussion to include these alternative causes with regard to the Pacific sector core ELT17-9:*

*"The Chaetoceros rs. group is associated with both WSI (Armand et al., 2005) and meltwater stratification (Crosta et al., 1997), and high abundances of Chaetoceros rs. in Ross Sea sediments deposited during past interglacial periods have been linked to increased upwelling and subsequent meltwater stratification within the Ross Sea Gyre (Kim et al., 2020). The high Chaetoceros rs.*

*abundance in core ELT17-9 during MIS 5e could therefore indicate an north-eastward shift of the Ross Sea Gyre from its modern day position (Dotto et al., 2018) and an accompanying displacement of meltwater circulation (Merino et al., 2016) and the WSI edge in the Pacific sector. It is also possible that the reduced Pacific sector WSIE during MIS 5e is associated with earlier seasonal sea-ice retreat during the austral spring and a longer open-ocean season, promoting a stronger spring bloom signal, of which the* Chaetoceros *group is a major component (Leventer, 1991)."*

6. Lines 383-391, discussion of changing ecosystem due to sea ice changes in the Weddell Sea sector: Consider references to:

Moline et al. 2004, Alteration of the food web along the Antarctic Peninsula in response to a warming trend. Global Change Biology. 10. 1973-1980 and Mendes et al., 2018, New insights on the dominance of cryptophytes in Antarctic coastal waters: A case study in Gerlache Strait. DSR II, 149, 161-170.

I realize that both these papers are from the western side of the Antarctic Peninsula but note the impact of meltwater and a warming ocean on primary producers and impacts that cascade through the food web.

We have expanded the discussion on ecosystem impacts to further detail the possibly effects of warming and meltwater release on the wider food web:

*"The sensitivity of Weddell Sea WSI to warmer climates could have substantial implications for the SO biosphere given the high rates of primary productivity in this region today (Vernet et al., 2019). Whilst a future reduction in WSIE and increase in glacial meltwater flux would be expected to promote primary productivity in the western part of the Weddell Sea (de Jong et al., 2012), the higher SSTs would not favour key trophic intermediaries, e.g. Antarctic krill (Euphausia superba) (Atkinson et al., 2017; Siegel and Watkins, 2016), and would therefore negatively affect megafauna at higher trophic levels (Hill et al., 2013). The impacts of warming and reduced WSIE on the SO food web are seen along the Antarctic Peninsula in the present day, with a recent shift in phytoplankton community structure from diatoms to smaller cryophytes, which are less efficiently grazed by Antarctic krill (Mendes et al., 2018; Moline et al., 2004). Future WSI edge retreat, at equivalent levels to MIS 5e, would also negatively impact upon modern sea-ice obligate species, such as Emperor and Adélie Penguins (Cimino et al., 2013; Jenouvrier et al., 2005)."*

7. Line 395 - remove parenthesis since this is an important consideration. In general, if it's important enough to be stated, no parentheses necessary.

We have removed the parentheses.

**Reviewer #2**

The manuscript of Chadwick et al., titled "Reconstructing Antarctic winter sea-ice extent during Marine Isotope Stage 5e", presents a comprehensive study of nine circum-Antarctic sediment records covering the time span of Marine Isotopic Stage 5e (130-116 ka BP) to reveal the temporal and spatial dynamic of Antarctic sea ice under warmer than present climate. The Manuscript (MS) is well written, logically structured, and addresses a major burning question in paleoclimatology at the moment, the behavior of the Southern Ocean environment in a time warmer than present. In addition to their earlier study of circum-Antarctic MIS 5e records based on diatom sea ice and surface temperature reconstructions, comprising previously published cores from locations north of the modern winter sea

ice boundary (Chadwick et al., 2020), this new study presents 7 out of 9 diatom records from locations south of the modern sea ice boundary. The authors highlight this fact as a major advantage, as all diatom-based studies before did present only records north of this significant boundary.

Such a study depends mostly on the reliability of the age models for the sediment core sections covering MIS 5e to get an appropriate synchronization of the sequence of events comprising Late Termination II (132-130 ka BP) and MIS 5e (130-116 ka BP) (see e.g. Bianchi and Gersonde, 2002). In the case of the present manuscript (MS), five out of nine chronologies for MIS 5e are based mainly on the correlation of downcore magnetic susceptibility measurements to a dated target curve, like the benthic foraminifera oxygen isotope curve of Lisiecki and Raymo (LR04; 2005). I don´t want to talk here about the known timing uncertainties of the benthic isotope stack especially in the Pacific sector of the Southern Ocean of several thousand years, but want to raise a problem concerning the use of magnetic susceptibility as an homolog for glacial-interglacial cyclicity in the circum-Antarctic seasonal ice zone. The theory behind chronologies based on magnetic susceptibility is the fact, that many studies before have shown increased concentrations of magnetizable particles in sediments related to glacial marine conditions, compared to sediments deposited during interglacial times. But most of these records come from open ocean sites and the glacial particle input is related to increased aeolian transport of iron-enriched dust during glacial times. However, for core sites in the seasonal sea ice zone around Antarctica this is different, because in some regions perennial sea ice cover might prevent dust particles from sinking down directly at the location where they hit the air-water interface and got displaced by sea ice movement. Furthermore, in the circum-Antarctic waters, especially in vicinity of the continent, we get another source of iron-rich particles, so-called ice rafted debris (IRD). IRD is transported mainly northward by ice bergs from the point of the Antarctic coast or shelve ice, where they calve. During glacial times, iceberg calving and thus northward transport of IRD might be reduced, compared to our present warm time or even times warmer than present, like MIS 5e. Thus, one needs to differentiate carefully between the different terrigenous particles in a sediment core, to exclude bias of the magnetic susceptibility due to IRD. The present MS fails to argue in that way, nor does it present a clarifying sedimentological analysis of these particles, and uses magnetic susceptibility-based chronologies without discussion, decreasing the reliability of these age models in my eyes significantly.

First of all, the chronologies for most of our studied cores are **not only** based on MS, and they also have been **previously published**, see Table 2 (*NB*: the age model for TPC287 is based on correlation of its MS record with that of neighbouring core TPC288, whose MS shows a very strong match with EDC dust and whose age model is additionally constrained by *Cycladophora davisiana* stratigraphy throughout the core and AMS [14]C dates in its upper part, see Pugh et al. (2009); AMS [14]C ages also constrain the upper part of TPC287, see Allen et al. (2011)). Furthermore, it is well documented that MS generally shows a positive correlation with foraminifera $\delta^{18}O$ stacks from MIS 1 to MIS 6 (as well as earlier glacial-interglacial cycles) **also south of the WSI edge,** with low MS during interglacials and usually high MS during glacials (cores with independent dating by AMS [14]C ages, *C. davisiana*, *Eucampia antarctica*, $\delta^{18}O$ foraminifera and/or biostratigraphy dating, relative paleointensity dating, tephra chronology), e.g. Scotia Sea (Pudsey & Howe 1998, Diekmann et al. 2000, Pugh et al. 2009, Allen et al. 2011, Xiao et al. 2016b), Antarctic Peninsula (Pudsey & Camerlenghi 1998, Pudsey 2000, Lucchi et al. 2002, Macrì et al. 2006, Yoon et al. 2009, Lee et al. 2012), Bellingshausen Sea (Hillenbrand et al. 2008, Hillenbrand et al. 2021), Amundsen Sea (Hillenbrand et al. 2009, Konfirst et al. 2012), offshore Prydz Bay (Wu et al. 2021), Weddell Sea-Lazarev Sea-Cosmonaut Sea (Bonn 1995) and Ross Sea (Kim et al. 2020a). The reason for this apparently circum-Antarctic pattern is the higher biogenic content (mainly diatoms, but also foraminifera and radiolarians) in interglacial sediments south of the Antarctic Polar Front (Grobe & Mackensen 1992), which is evident from palaeoproductivity proxies,

such as biogenic barium, opal and $CaCO_3$ contents and/or their accumulation rates (see all of the aforementioned papers but also Nürnberg et al. (1997); Bonn et al. (1998); Hillenbrand & Cortese (2006)). In contrast, contents and accumulation rates of IRD (gravel fraction) south of the modern WSI extent show a highly variable pattern throughout glacial-interglacial cycles (Diekmann et al. 2003) and thus cannot explain the consistent MS minima during interglacials.

While a very good correlation between MS records of marine sediment cores and the dust record of East Antarctic ice cores has been observed in several areas, including the Indian and Pacific sectors and especially the Scotia Sea, both north and south of the modern WSI edge (Petit et al. 1990, Diekmann et al. 2000, Thamban et al. 2005, Pugh et al. 2009, Mazaud et al. 2010, Weber et al. 2012, Xiao et al. 2016b), the reason for this correlation is still unclear. While some scientists suggested direct deposition of aeolian dust (e.g. Weber et al. 2012, Reviewer #2 here), other authors ruled out direct dust sedimentation and favoured transport controlled by ACC flow changes (Diekmann et al. 2000, Mazaud et al. 2010) or proposed early diagenetic processes, with biogenic magnetite formation in response to productivity changes caused by varying dust fertilization (Yamazaki & Ikehara 2012). More recently, based on analyses of sediment cores located south of the WSI limit around the Antarctic Peninsula, a dust-like MS signal has been attributed to changes in IRD input, mainly IRD in the coarse silt- to sand-sized fractions, by Kim et al. (2018), Kim et al. (2020b) and Shin et al. (2020).

In summary: (i) a very good MS-dust correlation **is not** restricted to the permanent open ocean zone, (ii) **its causality is not fully understood**, yet, and different factors may be responsible for it in different SO regions (the investigation of which, however, is far beyond the scope of our paper), and (iii) **MS patterns south of the WSI show similar patterns in numerous cores**, with MS minima always characterizing interglacial sediment intervals. This well documented pattern includes the cores studied by us, highlighting the usefulness of MS as a stratigraphic tool for establishing an initial age model, which, however (and as Reviewer #2 correctly points out), needs to be supported by independent age control in order to avoid misidentification of MS minima as MIS 5e that actually mark older interglacials or sometimes can occur in glacial-time sediments (e.g., see Fig. R2). Consequently, **the age models for our cores are not only based on MS but also other chronological constraints, most of which have been previously published in peer-reviewed journals** (Table 2). We are convinced that this is a valid scientific approach.

Furthermore, the authors of the MS failed to explain the causes leading to a significant lack of MIS 5e-covering sediment records south of the present winter sea ice extent. Interpretation of diatom records and the estimation of sea surface temperatures and sea ice concentration from diatom census via transfer functions are highly dependent of the preservation stage of diatom assemblages (Zielinski et al., 1998: Esper et al, 2010; Esper and Gersonde, 2014a,b). Significant dissolution biased the composition of assemblages from oceanic sites located in areas with about 75% or more WSI occurrence probability (Zielinski and Gersonde, 1997; Esper and Gersonde, 2014a). In general, these areas are characterized by low biogenic opal deposition (Geibert et al., 2005). As this is true for modern diatom assemblages, this might be even worse in sediment samples from glacial times, with an opal belt moving northward circumpolar and opal concentrations decreasing in the core sediments. Dissolution-biased downcore records of diatoms treated with Modern Analogue Technique-based transfer functions (TF) with preservation-adjusted reference data sets would lead than to a kind of non-analog situation in the analog sample selection sequence of the TF. Esper et al. (2010) have shown, that especially weakly silicified diatoms, like the sea ice diatoms Fragilariopsis curta and F. cylindrus are prone to selective dissolution, altering diatom assemblages in vicinity of sea ice to assemblages dominated by heavily silicified diatoms of intermediate temperature affinity and no sea ice relation (e.g. Fragilariopsis kerguelensis and Thalassiosira lentiginose). TF treatment of such

altered diatom assemblages leads to temperature overestimation and sea ice concentration underestimation (Esper and Gersonde, 2014a,b). This becomes obvious in the presented sea surface summer temperatures (Fig. 4), with maximum values of about 6°C in the late Termination II and >4°C during MIS 5e in most of the cores from the Atlantic and the Indian sectors, where sea surface summer temperatures of 0° to 1°C prevail today. Thus, the problem of selective diatom preservation might have led to the exclusion of many diatom records located south of the winter sea ice boundary from previous MIS 5e studies. However, in the MS of Chadwick et al., this issue is not addressed nor did the authors present any clue for the preservation stage of their nine diatom records. Neither did the authors present measurements of biogenic opal to proof the quality of the diatom assemblage concerning preservation.

We agree with the reviewer that the preservation of diatoms in marine sediments is an important consideration when running transfer function analysis and as such the dissolution of diatoms in all samples for this study were considered using the method detailed in Warnock et al. (2015). We checked the areolae in *F. kerguelensis* valves to ensure there was little or no expansion and conjoining, as would occur under a high degree of dissolution. We also checked that each analysed sample contained a mixture of both heavily and weakly silicified diatom valves over the whole size range, which was suggested by Zielinski (1993) as an indicator of good preservation. In the revised manuscript we have included additional text clarifying the importance of good preservation and the steps we have taken to check the MIS 5e diatom assemblages are suitably preserved for transfer function analysis:

*"For both the MAT and the FCC proxy, it is important that the diatom assemblage is well preserved, as high dissolution causes preferential loss of the more lightly silicified diatom species, generally sea-ice related species, and would therefore bias reconstructions towards warmer SSTs and lower sea-ice conditions. The samples used in this study were investigated for signs of dissolution following the procedure detailed in Warnock et al. (2015), whereby the areolae in* F. kerguelensis *valves were checked to ensure there was little, or no, expansion and conjoining, as would occur under a high degree of dissolution. Diatom assemblages in the analysed samples were also checked for a mixture of both heavily and weakly silicified diatoms across the whole size range, which was suggested by Zielinski (1993) as an indicator of good preservation. Poor preservation of diatoms in sediments located beneath heavy winter sea ice (SIC >75 %) has likely limited most previous attempts to reconstruct MIS 5e conditions from core sites located south of the modern mean WSIE, and thus the preservation of samples analysed in this study was carefully considered to avoid introducing a warm (low sea ice) bias into our reconstructions"*

It is also worth noting that, with the exception of core ANTA91-8, all of the cores in our study are located north or only slightly south of the modern 75 % WSI limit and would be expected to be north of this limit during a warmer-than-present interglacial. Therefore, the significant dissolution under >75 % WSI mentioned by the reviewer is not likely to be a concern for the majority of the cores analysed in this study. We reiterate that the present study focuses on the LIG and not glacial sediments.

To conclude, I want to highlight the scientific significance of the study presented by Chadwick et al. Addressing sea ice variability and temperature field changes in an environmental setting warmer than present day is very important for answering questions on the current climate change. However, the scientific approach chosen by Chadwick et al. needs a careful revision, corrections and a detailed consideration of diatoms as sea ice and temperature proxy in the seasonal sea ice zone around Antarctica. Besides the significant uncertainties I raised concerning the age models and the sea ice and temperature estimates based on diatom assemblages, the general form of presenting this MS is clear, concise and well-structured. Before getting this MS published, I highly recommend a detailed estimation of the preservation stage of each record to proof, that the quality of the assemblages is

good enough for a transfer function treatment. Such a quality appraisal could for example be done following the approach of Benz et al. (2016), who presented different levels for diatom preservation quality, TF estimate quality, and age model quality. Concerning age model construction, the authors may find some clues in the recent publication from Xiao et al., 2016, dealing with dating obstacles in the Atlantic sector of the Southern Ocean. In the current state I recommend a rejection of the MS until the authors have proven the applicability of diatoms as reliable environmental proxies in the seasonal sea ice zone and the reliability of the applied age models.

**General comments**

**Abstract**

1.) There is no information included in the Abstract, on which proxies (e.g. marine diatoms and TF-derived environmental conditions) the study is based. Nor is there any detail on methodology for sea ice and sea surface temperature reconstructions mentioned. However, both information would be of great value for getting the main idea behind the study immediately.

We have added in the Abstract further details on the proxies and methodology used in our study. It now reads:

*"Winter sea-ice extent and sea-surface temperatures are reconstructed using marine diatom assemblages and a Modern Analog Technique transfer function, and changes in these environmental variables between the three Southern Ocean sectors are investigated."*

2.) There is no proxy for meltwater flux or ACC flow band shifts mentioned in the Abstract.

We have added this information in the Abstract. It now reads:

*"High variability in the Atlantic sector winter sea-ice extent is attributed to high glacial meltwater flux in the Weddell Sea, indicated by increased abundances of the diatom species* Eucampia antarctica *and* Fragilariopsis cylindrus. *The high variability in the East Indian sector winter sea-ice extent is conversely believed to result from large latitudinal migrations of the flow bands of the Antarctic Circumpolar Current, inferred from latitudinal shifts in the sea-surface temperature isotherms"*

**Introduction**

3.) Line 057: Missing reference for a definition of the timing and length of MIS 5e – e.g. Fischer et al. (2018) define the Last Interglacial (LIG) (129-116 ka BP).

We have included a reference to Lisiecki & Raymo (2005).

4.) Line 075: Chadwick et al., 2020 previously presented a circum-Antarctic reconstruction of winter sea ice extent and sea surface temperatures. So what is the difference or gain of the new study?

The Chadwick et al. (2020) paper identified that, without more southerly records of sea ice during MIS 5e, the timing of the minimum sea-ice extent could only be constrained to an interval spanning 2-8 ka. This study presents more southerly records than Chadwick et al. (2020), predominantly south of the modern WSI limit, and is thus able to further constrain the timing of the minimum sea-ice extent during MIS 5e. Furthermore, we have identified two records (cores PC509 and ANTA91-8), which were covered by seasonal sea ice throughout MIS 5e – to our knowledge, the first such records reported in the literature. The use of more southerly core sites also allows this study to present a more detailed analysis of the patterns of sea-ice change between the three Southern Ocean sectors than was possible with the records analysed in Chadwick et al. (2020).

5.) Line 079: The main approach of this study, to transform qualitative sea ice extent and temperature variation estimates based on diatom assemblages into qualitative values of sea ice concentrations and sea surface temperatures is not described or referenced, concerning reliability and applicability in the working area.

We have included additional text clarifying the methods that will be used later in the study. It now reads:

*"This study presents new reconstructions of SO winter sea ice (WSI) during MIS 5e from the diatom assemblages preserved in nine marine sediment cores located south of 55 ºS and south of the modern Antarctic Polar Front (Figure 1). Qualitative reconstructions are based on the occurrence of sea-ice related diatoms (Gersonde and Zielinski, 2000). Quantitative estimates are produced through a diatom-based Modern Analog Technique transfer function, based on numerous core-top sediment samples (Figure 1) and originally detailed in Crosta et al. (1998). Quantitative and qualitative reconstructions of WSIE in the three SO sectors; the Atlantic sector (70 ºW – 20 ºE), the Indian sector (20 ºE – 150 ºE) and the Pacific sector (150 ºE – 70 ºW), are compared to answer the following questions:"*

**Material and Methods**

6.) Line 091: The question arises, why no diatom records from MIS 5e south of the modern winter sea ice extent have been published before. Are there factors hampering previous methods? How will these potential obstacles be overcome?

The reviewer's statement is incorrect as MIS 5e sea-ice extent south of the modern WSI limit has already been reconstructed for site PS2305-6 (Bianchi & Gersonde 2002). Nevertheless, we agree that there are probably only a few suitable target sites due to the issues highlighted by Reviewer #2 above and the fact that most of the Southern Ocean seafloor south of the modern WSI edge lies at water depths greater than 4000 metres, where sedimentation rates are low and chronologies difficult to develop. However, we have followed the strategy of Bianchi & Gersonde (2002). We predominantly checked for cores located just south of the modern WSI limit and assessed their suitability to provide a reliable MIS 5e sea-ice reconstruction.

7.) It is not mentioned how the effect of selective preservation of diatoms, especially south of the winter sea ice extent has been addressed! Bad preservation is a main factor negatively influencing transfer function results of diatom assemblages!

As mentioned above, we have included additional information on what steps we have taken to check and consider the preservation of the diatom assemblages analysed in our study. We have also included some text in response to point 6 highlighting how poor preservation may have hampered previous studies reconstructing MIS 5e sea ice:

*"For both the MAT and the FCC proxy, it is important that the diatom assemblage is well preserved, as high dissolution causes preferential loss of the more lightly silicified diatom species, generally sea-ice related species, and would therefore bias reconstructions towards warmer SSTs and lower sea-ice conditions. The samples used in this study were investigated for signs of dissolution following the procedure detailed in Warnock et al. (2015), whereby the areolae in* F. kerguelensis *valves were checked to ensure there was little, or no, expansion and conjoining, as would occur under a high degree of dissolution. Diatom assemblages in the analysed samples were also checked for a mixture of both heavily and weakly silicified diatoms across the whole size range, which was suggested by Zielinski (1993) as an indicator of good preservation. Poor preservation of diatoms in sediments located*

*beneath heavy winter sea ice (SIC >75 %) has likely limited most previous attempts to reconstruct MIS 5e conditions from core sites located south of the modern mean WSIE, and thus the preservation of samples analysed in this study was carefully considered to avoid introducing a warm (low sea ice) bias into our reconstructions"*

8.) Line 110: The reference of Crosta et al (2020) does not describe the application of MAT for sea ice reconstructions nor does it describe MAT in detail or deal with the mentioned D-257-33 configuration of the TF applied in the MS!

MAT257-33-5 (based on 257 reference samples, 33 taxa and up to 5 analogs) represents an evolution of the transfer function presented at length in Crosta et al. (1998); which was in its MAT195-33-5 configuration at that time. Over the last 20 years, new core-top samples have been added and many publications have been using incremented MAT approaches to reconstruct SST and/or sea ice, providing each time very robust results. Indeed, each time SST or sea-ice reconstructions agreed very well with other proxies produced in the same cores. To cite just a few studies for sea-ice reconstructions: Both Crosta et al. (2004) and Nielsen et al. (2004) used MAT201-31-5. In Crosta et al. (2004), the modern model was presented again and showed very similar results as in Crosta et al. (1998). Nair et al. (2019) and Ghadi et al. (2020) both used MAT249-33-5. A few studies for SST reconstructions: Both Shemesh et al. (2002) and Crosta et al. (2004) used MAT201-33-5, and Orme et al. (2020) and Civel-Mazens et al. (2021) used MAT249-33-5. Additionally, Ferry et al. (2015a) and Ferry et al. (2015b) demonstrated that a very different transfer function (GAM; an IKM-type based transfer function), which used only 4 diatom species as predictors and 163 core-top samples out of the 243 analogs composing the modern database at that time (only core-tops located at or south of the WSI edge), provided very similar down-core results for core SO136-111 as MAT201-33-5. Esper & Gersonde (2014a) also yielded similar results with IKM and MAT but chose MAT as it outcompeted the other transfer functions in terms of pure statistics (R2, slope, RMSEP). We also note that the quantitative sea-ice data produced by MAT204-5-33 for the Last Glacial Maximum (LGM) were in excellent agreement with those reconstructed from the FCC qualitative proxy in nearby LGM horizons, thus providing a robust picture of WSI cover for this time period (Gersonde et al. 2005). Finally, we note that the new transfer function reconstructions presented here are in very good agreement with both the FCC proxy for sea ice and the *A. tabularis* abundances for SSTs, as already evidenced in other cores (Nair et al. 2019, Ghadi et al. 2020).

In conclusion, we believe that MAT257-33-5 is a robust approach to quantitatively reconstruct sea-ice conditions in the Southern Ocean. It represents an incremental evolution of a database that has been robustly used over the last 20 years. We therefore deem it unnecessary to present again at length the performance of the transfer function besides its statistical values on the modern model. However, additional text has been added to clarify the evolution of the method from the version presented in Crosta et al. (1998):

*"The MAT257-33-5 (based on 257 reference samples, 33 taxa and up to 5 analogs) utilised in this study is an evolution of the MAT195-33-5 detailed in Crosta et al. (1998), with the addition of a further 62 surface sediment samples (Figure 1). The incremental evolutions of this transfer function over the last 20 years have yielded robust SST and sea-ice reconstructions when compared alongside other proxies within the same cores (e.g. Civel-Mazens et al., 2021; Crosta et al., 2004; Ghadi et al., 2020; Nair et al., 2019; Shemesh et al., 2002)."*

9.) Schweitzer (1995) is a bid old-fashioned for a sea ice reference data set (resolution only 2x2 deg) - see Esper and Gersonde (2014)

[Figure]

**Fig. R1:** Ensemble of five figures showing the sea-ice concentration data at one location (map 1) and at the four contiguous grid cells, one in each direction (north: map 2; east: map 3; south: map 4; west: map 5). Each move from the central pixel to the nearest pixel is less than 0.5°, as shown by the coordinates on the upper left of each map. Each grid cell presents different sea-ice concentration values. These two facts argue against the reviewer's statement that Schweitzer's numerical atlas represents mean sea ice climatologies on a 2*2° grid. The web page dedicated to the numerical atlas (https://geo-nsdi.er.usgs.gov/metadata/digital-data/27/metadata.faq.html) specifies that the 2*2° grid is another derivative product made for a specific program on the Pliocene.

We address the reviewer's question about grid resolution of the Schweitzer (1995) numerical atlas in Fig. R1.

The ice grid size varies depending on the region and channel. For the Southern Ocean, it varies from 25x25 km for the older detectors to 6.25x6.25 km for the most recent one (https://nsidc.org/data/polar-stereo/ps_grids.html). The projection is polar stereographic, with the origin of each x,y grid been the pole and projections been true at 70°S.

The extraction software ("look up ice") provided with the numerical atlas allows to interpolate to the surrounding pixels, resulting in a ~1*1° grid. Request can also be made for the exact pixel. Sea-ice concentration data extracted for several nearby core-tops from the Ross Sea and included in the modern database do present similar, but different, values. This again argues against the claim that Schweitzer's atlas is smoothed on a 2*2° grid.

The Schweitzer (1995) numerical atlas is based on SSMI data from 1978 to 1991. Ferry et al. (2015a) have shown that modern sea-ice cover is affected by human activities and that the most recent data should not be used as input data in a modern database when comparing core-top samples to modern data. This is because the core-top samples encompass several decades of sedimentation, but not the most recent years.

10.) Line 123: A relatively high uncertainty for sea ice concentration estimates!

Our RMSEP of 9% is in the same order of magnitude as the RMSEP of 6% in equivalent MAT transfer function studies (e.g. Esper & Gersonde 2014a).

11.) Line 124: As we have no information on the circum-Antarctic distribution of the TF reference samples, regional lacks for e.g. the Pacific sector cannot be addressed! It would therefore of great benefit to see the spatial distribution of the training data set of the TF to avoid regional biases.

We agree that the spatial distribution of the reference samples would be beneficial and have included them on the revised Figure 1.

**Age Models**

12.) Five out of nine sediment core chronologies for MIS 5e rely mostly on the comparison between magnetic susceptibility and the benthic foraminifera isotope stack of Lisiecki and Raymo (2005) (Table 2). This is problematic, because magnetic susceptibility records in the seasonal sea ice zone might be biased by ice rafted debris and seasonal ice cover, at the end not reflecting glacial-interglacial cyclicity. This becomes especially than problematic, if no other age source could be used in addition or comparison, like in cores TPC287, NBP9802-04, and ANTA91-8. Totally questionable is the dating method for core PC509, using wet bulk density as a proxy for biogenic opal, which is a proxy for glacial-interglacial productivity changes. Thus, this parameter is prone to several alteration processes, starting with changing downcore sedimentation rates leading to different compaction rates and not ending with selective diatom preservation altering the opal content. Thus measurements of magnetic susceptibility or biogenic opal are good proxies for a quick and dirty age model, especially onboard a research vessel, but lack the reliability needed for timing the climatic events related to MIS 5e to be presented in a research paper.

[Figure]

**Fig. R2:** Correlation of physical properties and Ba/Rb records (plotted as log-normalised [LN] peak-area ratios following Weltje & Tjallingii (2008)) between cores PC509 and PC723/GBC724, a 11.17 m long core dating back to MIS 8 (Hillenbrand et al. 2021). Depths are given in centimetres composite depth (cmcd) for core

PC723/GBC724, a spliced record of piston core PC723 with box core GBC724 collected from the same site, and in centimetres below seafloor (cmbsf) for core PC509. Correlations are marked by dashed dark red lines, and the MIS 5e interval is highlighted by grey shading. Whole-core volume MS of core PC723/GBC724 was measured at 2.5 cm depth resolution with a loop sensor. Wet-bulk density of PC723/GBC724 as well as the whole-core MS and wet-bulk density records of PC509 were measured at 1 cm depth resolution with a multi-sensor core logger. Ba/Rb ratios were analysed with an Avaatech XRF-core scanner at 0.5 cm depth resolution for PC723/GBC724 and 0.25 cm resolution for core PC509. The MIS 5 interval in PC723/GBC724 was identified by planktic foraminifera $\delta^{18}O$ stratigraphy (see Hillenbrand et al. 2021).

We have explained above why we believe that MS can be used to construct the age models for our cores. With regards to PC509, the age model has been established by correlating the core's wet-bulk density with the LR04 stack. In marine sediment cores, downcore changes in (wet-bulk) density often mirror those of biogenic opal content (Weber et al. 1997), and in sediment cores retrieved south of the Antarctic Polar Front opal contents typically vary on glacial-interglacial timescales (Bonn 1995, Hillenbrand et al. 2009). The validity of our dating approach for MIS 5e is justified by the correlation of the wet-bulk density, MS and especially the Ba/Rb records of core PC509 with those of core PC723/GBC724 from the Antarctic continental rise in the southern Bellingshausen Sea (Fig. R2). The Ba/Rb ratio is an indicator for biogenic barium, the most reliable palaeoproductivity proxy in sediment cores from south of the Polar Front (Nürnberg et al. 1997, Bonn et al. 1998, Hillenbrand & Cortese 2006, Jaccard et al. 2013). The age model for core PC723/GBC724 has been published by Hillenbrand et al. (2021), with the MIS 5e interval being identified by planktic foraminifera $\delta^{18}O$ data.

13.) I wonder, why the reliable diatom stratigraphic marker Rouxia leventerae (Zielinski et al., 2002) has not be applied, as detailed diatom assemblage should be available for this study. I also wonder, why the diatom stratigraphic marker Hemidiscus karstenii has been used, although this diatom got extinct end of MIS 7 (about 191 ka BP according to Zielinski and Gersonde, 2002). The biostratigraphic approach needs to be improved.

All of the diatom assemblages sampled in this study have <1 % *Rouxia leventerae* and are therefore all younger than the ~135 ka LOD identified by Zielinski et al. (2002). The *H. karstenii* stratigraphic marker was used to ensure that the identified interglacial period in the MS record was younger than MIS 7. Additional text has been added to clarify how the *R. leventerae* abundance further constrains our age models:

*"These published chronologies are further constrained by checking the abundance of the diatom species* Rouxia leventerae *in all MIS 5e samples. All diatom assemblages analysed in this study have* R. leventerae *abundances <1 %, which suggest that the considered sediments are younger than the ~135 ka Last Occurrence Datum identified by Zielinski et al. (2002)."*

14.) The age uncertainties of all cores are >2.5 ka, projecting discussions of leads and lacks of sea ice processes compared in different Antarctic sectors into the error range! Tuning only one proxy record (e.g. magnetic susceptibility) to a target curve (e.g. oxygen isotopes) for each core is a bit weak. It would be good to have at least one biostratigraphic datum for each core to get a starting point for the tuning correlations.

We agree that these age uncertainties are fairly large and thus a large focus of this paper is on the different patterns of sea-ice change across the three Southern Ocean sectors rather than interrogating short leads and lags between sectors. Furthermore, we point out that in previous studies other authors developed age models for cores spanning MIS 1-6 "based on a correlation of physical parameters, XRF-derived elemental composition (e.g. Fe counts), diatom assemblage composition and derived sea ice and sea surface temperature with the EDC [=EPICA Dome C] ice core record and diatom

biostratigraphic data" (e.g. Esper & Gersonde 2014a). Given the different hypotheses about the exact reason for the match between MS/iron concentrations in some Southern Ocean sediment cores and dust concentrations in Antarctic ice cores (see above) and the fact that synchrony between air and sea surface temperature peaks/changes is merely an assumption, also such age models inherently bear uncertainties that can render identification of leads and lags impossible, even though this is rarely mentioned in corresponding publications (for an exception see Pugh et al. (2009)). In this regard we also want to point out that even Antarctic ice core chronologies can have (at least initially) age uncertainties of a few thousand years during MIS 5. For example, Narcisi et al. (2006) identified a tephra layer in a horizon of the EDC ice core which had been assigned an age of 86.7 ka on the EDC2 age scale. This tephra layer had previously been $^{40}Ar/^{39}Ar$ dated near its source volcano in Marie Byrd Land (West Antarctica) to 92.0 ± 0.9 ka and 92.5 ± 0.9 ka, respectively (Wilch et al. 1999). This $^{40}Ar/^{39}Ar$ tephra age was subsequently used for the EDC3 age scale by Parrenin et al. (2007).

15.) Line 162-166: The discussion on the use of diatom of the genus Rouxia lacks necessary details. First, one should use the presence or absence of the defined species Rouxia leventerae only, as e.g. Rouxia constricta got extinct end of MIS 8 (about 280 ka ago) and its presence would point to significant reworking. Second, an abundance of R. leventerae > 1% for MIS 6 diatom assemblages is reported from core location north of the present winter sea ice edge only (Zielinski and Gersonde, 2002), not neglecting possible influence of selective preservation on the record ANTA91-8 far south of this boundary and therefore altering the maximum abundance of R. leventerae!. Thus, without a detailed examination of the diatom preservation, Rouxia sp. does not corroborate anything.

The *Rouxia* spp. abundances discussed in this study only include *R. leventerae* and this is clarified in the revised manuscript. Previous studies have utilised *R. leventerae* as a stratigraphic marker in regions south of the WSIE, both on the Ross Sea continental shelf (Bart et al. 2011) and the Wilkes Land margin (Jimenez-Espejo et al. 2020), and so we believe it is justified to use it for core ANTA91-8 in this study. It should also be noted that *R. leventerae* is a heavily silicified diatom species, for which relative abundances would increase if the assemblage was subject to strong dissolution. *Rouxia leventerae* abundances <1 % during MIS 5e therefore argue against selective preservation in this core.

**Results**

16.) First of all, it is important to remark, that not all cores exhibit the same chronological resolution. Core U1361A for example, has only 6 samples within 12ka, leading to a resolution of one sample per 2k years. Other cores, like TPC287 and ANTA91-8 have a better resolution of one sample per 800 years. Only the latter cores are than appropriate to indicate short variations in sea ice cover and surface temperatures. Low res cores are prone to signal distortion due to uncertainty changes.

We agree with the reviewer that low chronological resolution inhibits our ability to identify short term variations in sea ice and SSTs, and this point has been discussed with reference to the record from Hole U1361A in lines 368-374 and 444-448.

17.) MAT sea ice concentration estimates and FCC cumulative abundances indicate for the Atlantic sector low to intermediate sea ice cover during the late glacial stage 6, increased sea ice values during Termination II and relatively high sea ice cover during MIS 5e (Figure 3). Sea ice records for the Indian and Pacific sectors indicate low, but constant sea ice cover over the whole analyzed interval (132-120 ka). Taking into account, that only the two Atlantic cores TPC290 and TPC288 are located north of the modern winter sea ice edge, the WSIc values at least for the glacial Termination are far too low, indicating a sea ice retraction from the modern winter extension at most of the core locations during the glacial and sea ice expansion during MIS5e (especially at core position TPC287).

The low sea-ice cover between 131-130 ka in the three Atlantic sector records is within chronological uncertainty of the Antarctic ice core thermal optimum at 129-128 ka, and the re-expansion of sea ice ~4 ka after this minimum is consistent with the pattern and relative timings in core PS2305-6 published by Bianchi & Gersonde (2002) (as discussed in lines 305-307). The low sea-ice cover in our records before 130 ka may be due to a chronological offset, with the sea-ice minimum in these cores actually occurring at 129-128 ka. However, the signals could also be real, and an early retreat of sea ice during glacial Termination II is consistent with results of Bianchi & Gersonde (2002) showing that in nearly all of their studied cores, such as PS2305-6 (see Fig. 1 in the main manuscript), FCC abundances fell below 3% before or at 130 ka. Furthermore, the model experiments of Menviel et al. (2010) demonstrated that during early MIS 5e the release of vast quantities of glacial meltwater into the surface ocean caused by Antarctic ice-sheet loss, especially the potentially major deglaciation affecting the West Antarctic Ice Sheet (WAIS), would have caused SST reduction and sea-ice expansion. Importantly, this meltwater injection into the surface waters of the Antarctic Zone, which is supported by the observation of meltwater "spikes" characterizing planktic foraminifera $\delta^{18}O$ data in cores from the Weddell Sea continental margin during glacial-interglacial transitions (Grobe et al., 1990), would also have resulted in a warming of subsurface waters that, in turn, would have led to further ocean-forced melting of the ice-sheet grounding zones, especially of the marine-based WAIS, and triggered a positive feedback loop (Menviel et al., 2010; Bronselaer et al., 2018). Because of their location near the centre of "Iceberg Alley" (Weber et al., 2014), cores TPC287 and TPC288 can be expected to be particularly sensitive for recording such meltwater events. Therefore, we believe that re-tuning the chronology of our records to make the minimum sea-ice concentration synchronous with the peak Antarctic air temperatures in the ice core records makes too many assumptions (see also above) and would "cover up" a potentially important early retreat of sea ice in the Atlantic sector with subsequent expansion due to glacial meltwater release.

18.) Line 196: Unfortunately, the reference of Armand et al., 2005, assuming F. separanda to be related with sea ice is not state of the art any more. Esper et al. (2010) indicate a wide temperature range of -0.5°C to 4°C for this species in marine sediments. Surface water studies report a temperature range of 1.2°C to 8.7°C, which makes it unlikely that F. separanda is a typical sea ice related taxon (Esper et al., 2010). Esper and Gersonde (2014b) show a temperature range of F. separande in 336 surface sediments between 0° to 8°C with a maximum abundance occurring around 2°C. Thus, the presented explanation of F. separanda biasing the TF to colder temperature values and higher sea ice concentrations in not likely.

We thank the reviewer for bringing this to our attention and have rewritten the text to discuss why the location of the single analog for this sample indicates that it is unlikely to be a representative modern analog:

*"For this sample, only a single modern analog could be identified, indicating that the fossil diatom assemblage is different from almost everything in the modern reference database. The single selected analog is not chosen by the transfer function for any of the other MIS 5e samples from core TPC287, indicating that it is unlikely to be a truly representative modern analog for the MIS 5e condition at this core site. The location of this single selected analog, which is further south than any of the analogs chosen for the other MIS 5e samples from core TPC287, suggests that the fossil assemblage has been biased towards colder, heavier sea-ice conditions, probably due to dissolution or transport of the preserved assemblage."*

19.) Regarding the reliability of sea ice TF results, one should not rely on the dissimilarity threshold only, but should have a look to the origin of the 5 analogs itself. As a quality measure, one can say, that the closer the analogs were selected regarding the core location, the more likely the relationship

between assemblage composition and target environmental variable is. Esper and Gersonde (2014) showed significant differences of the diatom assemblage composition in the three Antarctic sectors regarding locations in sea ice vicinity. Thus, it could be important for a reliable sea ice concentration estimate, that the MAT analogs are selected as close as possible to the core location, or at least coming from the same Antarctic sector.

We have included a discussion of the locations for our analogs:

*"To check for other potentially anomalous palaeo-reconstructions, the number of times each modern reference sample was selected as an analog were considered (Supplementary Figure 2). Fossil samples were separated into three MIS 5e-Termination II time intervals (following the approach of Chadwick et al. (2022)) and modern reference samples that are only selected as analogs for a small number (<5) of fossil samples were identified (Supplementary Figure 2). None of these less-selected reference samples are the primary or sole analog for an MIS 5e fossil sample and are therefore unlikely to result in an unrepresentative Sept. SIC (or SSST) reconstruction."*

It is not possible to produce a map of analog locations for each individual sample but we have included maps with the selected analogs for specific time intervals (e.g. the 132-130 ka Termination II interval) in the supplementary material.

20.) Generally, the authors seem to avoid to describe their signal in full chronological length, starting at the end of stage 6. Otherwise, I am not able to explain, why they do not question the low sea ice values during the glacial Termination II. For example, Atlantic core TPC287 indicates low sea ice values before 130 ka, highest sea ice concentration at the Antarctic Ice Core Thermal Optimum around 129 ka and higher than Termination II sea ice values across MIS 5e. A similar pattern can be found in the Pacific sector. This is in direct contrast to the results published by Bianchi & Gersonde (2002) for the area directly north of the modern sea ice edge in the Atlantic sector.

We have addressed this comment in our response to point 17 and have clarified in the revised manuscript the possible reasons behind such low sea ice values during glacial Termination II:

*"We cannot rule out that the apparent retreat in Atlantic sector sea ice to a minimum during Termination II followed by a sea-ice expansion coincident with peak Antarctic air temperatures is an artefact caused by chronological uncertainties, with the WSIE minimum actually occurring alongside the peak Antarctic air temperatures at ~128 ± 1.5 ka (Holloway et al., 2017; Parrenin et al., 2013a). However, a genuine early (i.e., before 130 ka) retreat in Atlantic sector sea ice would also be consistent with most of the Termination II and MIS 5e records from this sector analysed by Bianchi and Gersonde (2002). Model experiments by Menviel et al. (2010) have demonstrated that during early MIS 5e the release of vast quantities of glacial meltwater into the surface waters of the Antarctic Zone (i.e., the region south of the Antarctic Polar Front) caused by Antarctic ice sheet deglaciation, especially the potential partial or total loss of the West Antarctic Ice Sheet (WAIS), would have led to SST reduction and equatorward sea-ice expansion. Importantly, this meltwater injection into the SO, which is supported by the observation of meltwater "spikes" characterizing planktic foraminifera $\delta^{18}O$ data in cores from the Weddell Sea continental margin during glacial-interglacial transitions (Grobe et al., 1990), would also have resulted in a warming of subsurface waters that, in turn, would have triggered further ocean-forced melting of the ice-sheet grounding zones, especially of the predominantly marine-based WAIS, thus kick starting a positive feedback loop (Bronselaer et al., 2018; Menviel et al., 2010). Because of their location within "Iceberg Alley", a main pathway of Antarctic icebergs travelling with the clockwise Weddell Gyre from the southern Weddell Sea Embayment into the Scotia Sea (Weber et*

*al., 2014), core TPC290 and especially cores TPC287 and TPC288 can be expected to be particularly sensitive for recording such meltwater supply."*

21.) In the Atlantic sector, cores TPC288 and TPC287 show highest sea surface temperature during glacial Termination II, a significant drop in temperature during the Antarctic Ice Core Thermal Optimum and lower than glacial temperature during the Last Interglacial!?! For example, core TPC287, today located in the seasonal sea ice zone with modern summer temperatures around 0°C (Fig. 1) shows post-glacial (!) temperatures of 6°C and interglacial values below 1°C. Core TPC288 slightly north positioned from that shows also 6°C post-glacial and up to 4°C interglacial temperatures, but in the Antarctic Ice Core Thermal Optimum interval, temperatures drop to nearly 0°C. What´s going on there?

As discussed in our responses to point 17 and 20, these high SSTs during the glacial termination are within chronological uncertainty of the thermal optimum in Antarctic ice cores. As with the sea-ice values, we cannot rule out a chronological offset as explanation of the high temperatures ~2 ka earlier than 'expected' and the subsequent cold temperatures during the thermal optimum, which we have mentioned in the revised manuscript:

*"Atlantic sector SSSTs reach their maxima during Termination II before a substantial drop coincident with the peak Antarctic air temperatures in ice cores (Parrenin et al., 2013a). As with the down-core Sept. SIC profiles, this offset may result from chronological uncertainties, with the highest SSSTs actually occurring alongside peak Antarctic air temperatures at ~128 ± 1.5 ka. However, air temperature and SST reconstructions from the Antarctic Peninsula and Scotia Sea have shown that during Termination I temperatures peaked at higher values than during the Holocene (Mulvaney et al., 2012; Xiao et al., 2016), thus, our records could indicate an equivalent early warming during Termination II for this region. Also, if the high air temperatures at ~128 ± 1.5 ka caused substantial Antarctic ice sheet loss, then the cold SSSTs in our ice-sheet proximal records at this time could, as discussed above, actually reflect major input of cold and fresh meltwater not recorded in cores further north."*

It should also be noted that air temperature reconstructions from the James Ross Island ice core in the Antarctic Peninsula record higher than modern temperatures during glacial Termination I (Mulvaney et al. 2012). Similarly, many marine records display higher SSTs during glacial Termination I than during the Holocene (Xiao et al. 2016a). Also, we consider that, if high air temperatures during the Antarctic **Ice Core** Thermal Optimum caused significant loss of glacial ice from the Antarctic Ice Sheet, our ice-sheet proximal core records could have picked up major input of cold and fresh meltwater not recorded in cores further north.

22.) The authors support their MAT-based temperatures with so-called "subtropical diatom" abundances, which have not been defined as a group or even mentioned in the Material & Methods section. Neither do they present counts or graphs of single diatom species, which could help to identify the nebulose "subtropical" species. The subtropical species within the Romero et al. (2005) reference have their habitat indeed in the subtropical zone, about 20° in latitude to the north of the core locations, with water temperatures of >10°. It is obvious, that those species are rather unlikely to be endemic in the Antarctic Zone, thus their sporadic appearance might be addressed to lateral transport. On the other hand, the term "subtropical" might be misleading. I recommend a table of those species belonging to the group. Assuming truly "subtropical" species to be present in the core, this would also bias the TF estimates by shifting the "collection area" of the 5 analogs too far to the north, leading to higher surface temperature averages and low sea ice concentrations.

Of the diatom species/groups presented in Romero et al. (2005), the dominant one in our samples is *Azpeitia tabularis*, which is associated with warmer waters north of the Antarctic Polar Front but also occurs in the Antarctic Zone in low abundances, without evidence for lateral current transport from further north. We have amended the figures and text to include only *A. tabularis* abundances as a warm-water proxy for this study:

*"The relative abundance of the diatom species Azpeitia tabularis is used as a comparison with reconstructed SSSTs. Azpeitia tabularis is a warm water species restricted to the region north of the maximum WSIE (Zielinski and Gersonde, 1997), with abundances <5 % in surface sediments south of the modern Antarctic Polar Front (Esper et al., 2010; Romero et al., 2005). Increasing abundances of this species in high latitude SO sediments therefore indicate warmer SSTs and ice-free conditions."*

**Discussion**

23.) Generally, it makes no sense to comment on the **Discussion** section in detail at this point, as changes in the environmental variables to be discussed may change significantly, if all the questions raised concerning diatom assemblage reliability and age model reliability have been addressed. In the following, I will comment therefore only on issues independent from age and sea ice/temperature estimates.

24.) Line 272-272: First of all, according to Crosta et al. 1997, increased Chaetoceros Resting Spore (CRS) abundance might point to higher meltwater discharge from Antarctica. Such a CRS peak occurs in many known diatom records around Antarctica within glacial terminations II and I (e.g. Bianchi and Gersonde, 2002, 2004; Benz et al, 2016). Second, I wonder why CRS are included in the TF. According to multivariate statistical analyses (e.g. Esper et al., 2010; Esper and Gersonde, 2014a,b) CRS variance is neither related to sea surface temperature variability nor sea ice cover variability. Esper and Gersonde (2014) discuss the unnecessary integration of reference species not related to the variance of the target variables (temperature, sea ice). I suggest to adjust the TF reference data set to avoid such phenomena.

We agree with the reviewer that the CRS abundance may be related to higher meltwater flux during Termination II or MIS 5e and have incorporated this information in the revised manuscript. The discussion of CRS abundances in lines 272-277 was not intended to explain the cause of the higher CRS abundances in core TPC290 but to mention the implications that these high abundances may have on the location of analogs chosen by the transfer function:

*"The discrepancy between Sept. SICs and FCC relative abundances at ~127 ± 2.6 ka in core TPC290 (Figure 3) is likely due to increased* Chaetoceros *resting spore (rs.) abundance at this time (Chadwick and Allen, 2021h). This* Chaetoceros *rs. abundance increase is also observed in the nearby core PS2305-6 (Bianchi and Gersonde, 2002) and is inferred to be caused by higher meltwater and iceberg flux at this time (Bianchi and Gersonde, 2002; Crosta et al., 1997). For core TPC290, there is a scarcity of modern analogs from the Scotia Sea region (Figure 1) and thus, the high* Chaetoceros *rs. abundances in MIS 5e samples are associated with modern analogs from sites along the Antarctic Peninsula, where SICs are greater than in the Scotia Sea."*

With regards to the inclusion of CRS in the transfer function, we concur that this group is not directly associated with either SSTs or sea-ice conditions, but, in the modern database, high CRS abundances are generally limited to SSTs <2 °C and Sept. sea-ice concentrations >60 % (Armand et al. 2005), thus providing a cold, heavy sea-ice end-member to the transfer function. Furthermore, the removal of CRS abundances from samples, such as those in core PC509, where CRS abundances are >70 % throughout MIS 5e, would result in the comparison of species % abundances estimated from <100 valves with the

surface samples, if they are not scaled to CRS-free counts. However, rescaling the accompanying diatom species to CRS-free counts is a real issue that can also bias the transfer function. For example, surface sediments from the Antarctic Peninsula margin, where SST is 1-3 °C and WSI is 40-70 %, often contain diatom assemblages dominated by CRS >70 %. In this case, *F. kerguelensis* accounts for 5-10 %. Rescaling to CRS-free counts will artificially increase the occurrence of *F. kerguelensis* to 40-50 %, i.e. similar relative abundances to the ones found in the open ocean, where SST is much higher and WSI much lower. In these open ocean samples, CRS are almost absent and rescaling will not change the original *F. kerguelensis %*. Overall, this will inject a lot of noise, with *F. kerguelensis* (again an example species) having similar relative abundances under very different environmental conditions. If rescaling was applied in Esper & Gersonde (2014b) or Esper & Gersonde (2014a) (information not provided in these publications), it may explain that *F. kerguelensis* is quite close to the CCA centre, as is CRS, and bears little environmental information in that database.

We stress that the MAT is a comparative approach that is neither based on the calculation of factors nor a paleo-ecological equation, such as IKM or WA-PLS. It is much less impacted by the presence of species with a low relationship to the target parameter. This is demonstrated by the fact that MAT201-33-5 provided very similar results to GAM, the latter using only 4 species (Ferry et al. 2015a, Ferry et al. 2015b).

To conclude, we believe that removing CRS from the core-top database without a sensible way to cope with the above mentioned issues of under-representation of accompanying species or of rescaling to CRS-free counts, two approaches criticised by pure statisticians, would substantially reduce the robustness of the transfer function.

**Conclusions**

25.) In general, the **Conclusions** section continues the **Discussion** section by adding more comparisons between own results and the literature. In my opinion, the **Conlusions** section should be reduced to a short recapitulation of the major results and significant statements on the implications. The remaining text could be integrated into the **Discussion** section.

We have restructured the conclusions and discussion as suggested by the reviewer.

**References**

Benz, V., Esper, O., Gersonde, R., Lamy, F., Tiedemann, R. (2016): Last Glacial Maximum sea surface temperature and sea-ice extent in the Pacific sector of the Southern Ocean. Qua-ternary Science Reviews 146, 216-237.

Bianchi, C., Gersonde, R. (2002): The Southern Ocean surface between Marine Isotope Stages 6 and 5d: Shape and timing of climate changes. Palaeogeography, Palaeoclimatology, Palaeoecology, 187, 151-177.

Bianchi, C., Gersonde, R. (2004): Climate evolution at the last deglaciation: the role of the Southern Ocean. Earth and Planetary Sciece Letters 228 (3), 407-424.

Esper, O., Gersonde, R., Kadagies, N. (2010): Diatom distribution in southeastern Pacific surface sediments and their relationship to modern environmental variables. Paleogeography, Paleoclimatology, Paleoecology 287, 1-27. Doi:10.1016/j.palaeo.2009.12.006.

Esper, O., Gersonde, R. (2014a): New tools for the reconstruction of Pleistocene Antarctic sea ice, Palaeogeography, Palaeoclimatology, Palaeoecology, 399, 260-283.

Esper, O., Gersonde, R. (2014b): Quaternary surface water temperature estimations: New diatom transfer functions for the Southern Ocean. Palaeogeography, Palaeoclimatology, Palaeoe-cology, 414, 1-19. Doi:10.1016/j.palaeo.2014.08.008.

Romero, O., Armand, L.K., Crosta, X., Pichon, J.-J. (2005): The biogeography of major diatom taxa in Southern Ocean sediments: 3. Tropical/Subtropical species. Palaeogeogr., Palaeoclimatol., Palaeoecol. 223, 49-65.

Xiao, W., Frederichs, T., Gersonde, R., Kuhn, G., Esper, O., Zhang, X. (2016): Constraining the dating of late Quaternary marine sediment records from the Scotia Sea (Southern Ocean). Quaternary Geochronology 31, 97-118.

Zielinski, U., Gersonde, R. (1997): Diatom distribution in Southern Ocean surface sediments (Atlantic sector): implications for paleoenvironmental reconstructions. Palaeogeogr. Palaeocli-matol. Palaeoecol. 129, 213–250.

Zielinski, U., Gersonde, R., Sieger, R., Fütterer, D. (1998): Quaternary surface water temperature estimations: Calibration of a diatom transfer function for the Southern Ocean. Paleoceanogr. 13 (4), 365-383.

Zielinski, U., Gersonde, R. (2002): Plio-Pleistocene diatom biostratigraphy from ODP Leg 177. Atlantic sector of the Southern Ocean. Marine Micropaleontology, 45:225-268.

Zielinski, U., Bianchi, C., Gersonde, R., Kunz-Pirrung, M. (2002): Last occurrence datums of the diatoms Rouxia leventerae and Rouxia constricta: indicators for marine isotope stages 6 and 8 in Southern Ocean sediments, Marine Micropaleontology, 46, 127-137.

**Rebuttal references:**

Allen C.S., Pike J. & Pudsey C.J. 2011. Last glacial–interglacial sea-ice cover in the SW Atlantic and its potential role in global deglaciation. *Quaternary Science Reviews*, **30** (19-20): 2446-2458.

Armand L.K., Crosta X., Romero O. & Pichon J.-J. 2005. The biogeography of major diatom taxa in Southern Ocean sediments: 1. Sea ice related species. *Palaeogeography, Palaeoclimatology, Palaeoecology*, **223** (1-2): 93-126.

Bart P.J., Sjunneskog C. & Chow J.M. 2011. Piston-core based biostratigraphic constraints on Pleistocene oscillations of the West Antarctic Ice Sheet in western Ross Sea between North Basin and AND-1B drill site. *Marine Geology*, **289** (1-4): 86-99.

Bianchi C. & Gersonde R. 2002. The Southern Ocean surface between Marine Isotope Stages 6 and 5d: Shape and timing of climate changes. *Palaeogeography, Palaeoclimatology, Palaeoecology*, **187**: 151-177.

Bonn W.J. 1995. Biogenic opal and barium: Indicators for late Quaternary changes in productivity at the Antarctic continental margin, Atlantic Sector. *Reports on Polar and Marine Research*, **180**: 1-186.

Bonn W.J., Gingele F.X., Grobe H., Mackensen A. & Futterer D. 1998. Palaeoproductivity at the Antarctic continental margin: opal and barium records for the last 400 ka. *Palaeogeography, Palaeoclimatology, Palaeoecology*, **139**: 195-211.

Bronselaer, B., Winton, M., Griffies, S.M., Hurlin, W.J., Rodgers, K.B., Sergienko, O. V., et al. 2018. Change in future climate due to Antarctic meltwater. *Nature*, **564** (7734): 53-58.

Chadwick M., Allen C.S., Sime L.C., Crosta, X. & Hillenbrand C.-D. 2022. How does the Southern Ocean palaeoenvironment during Marine Isotope Stage 5e compare to the modern? *Marine Micropaleontology,* **170**: 102066.

Chadwick M., Allen C.S., Sime L.C. & Hillenbrand C.D. 2020. Analysing the timing of peak warming and minimum winter sea-ice extent in the Southern Ocean during MIS 5e. *Quaternary Science Reviews*, **229**: 106134.

Civel-Mazens M., Crosta X., Cortese G., Michel E., Mazaud A., Ther O., Ikehara M. & Itaki T. 2021. Antarctic Polar Front migrations in the Kerguelen Plateau region, Southern Ocean, over the past 360 kyrs. *Global and Planetary Change*, **202**: 103526.

Crosta X., Pichon J.J. & Burckle L.H. 1998. Application of modern analog technique to marine Antarctic diatoms: Reconstruction of maximum sea-ice extent at the Last Glacial Maximum. *Paleoceanography*, **13** (3): 284-297.

Crosta X., Sturm A., Armand L. & Pichon J.-J. 2004. Late Quaternary sea ice history in the Indian sector of the Southern Ocean as recorded by diatom assemblages. *Marine Micropaleontology*, **50** (3-4): 209-223.

Diekmann B., Futterer D., Grobe H., Hillenbrand C.-D., Kuhn G., Michels K., Petschick R. & Pirrung M. 2003. Terrigenous Sediment Supply in the Polar to Temperate South Atlantic: Land-Ocean Links of Environmental Changes during the Late Quaternary. In: *The South Atlantic in the Late Quaternary: Reconstruction of Material Budgets and Current Systems*, Wefer G., Mulitza S. & Ratmeyer V. Eds. Springer-Verlag Berlin**:** 375-399.

Diekmann B., Kuhn G., Rachold V., Abelmann A., Brathauer U., Futterer D., Gersonde R. & Grobe H. 2000. Terrigenous sediment supply in the Scotia Sea (Southern Ocean): response to Late Quaternary ice dynamics in Patagonia and on the Antarctic Peninsula. *Palaeogeography, Palaeoclimatology, Palaeoecology*, **162**: 357-387.

Esper O. & Gersonde R. 2014a. New tools for the reconstruction of Pleistocene Antarctic sea ice. *Palaeogeography, Palaeoclimatology, Palaeoecology*, **399**: 260-283.

Esper O. & Gersonde R. 2014b. Quaternary surface water temperature estimations: New diatom transfer functions for the Southern Ocean. *Palaeogeography, Palaeoclimatology, Palaeoecology*, **414**: 1-19.

Ferry A.J., Crosta X., Quilty P.G., Fink D., Howard W. & Armand L.K. 2015a. First records of winter sea ice concentration in the southwest Pacific sector of the Southern Ocean. *Paleoceanography*, **30** (11): 1525-1539.

Ferry A.J., Prvan T., Jersky B., Crosta X. & Armand L.K. 2015b. Statistical modeling of Southern Ocean marine diatom proxy and winter sea ice data: Model comparison and developments. *Progress in Oceanography*, **131**: 100-112.

Gersonde R., Crosta X., Abelmann A. & Armand L. 2005. Sea-surface temperature and sea ice distribution of the Southern Ocean at the EPILOG Last Glacial Maximum—a circum-Antarctic view based on siliceous microfossil records. *Quaternary Science Reviews*, **24** (7-9): 869-896.

Ghadi P., Nair A., Crosta X., Mohan R., Manoj M.C. & Meloth T. 2020. Antarctic sea-ice and palaeoproductivity variation over the last 156,000 years in the Indian sector of Southern Ocean. *Marine Micropaleontology*, **160**: 101894.

Grobe H. & Mackensen A. 1992. Late Quaternary Climatic Cycles as Recorded in Sediments from the Antarctic Continental Margin. In: *The Antarctic Paleoenvironment: A Perspective on Global Change: Part One, Volume 56*, Kennett J.P. & Warkne D.A. Eds., Antarctic Research Series.

Grobe, H., Mackensen, A., Hubberten, H.-W., Spiess, V. & Fütterer D.K., 1990. Stable isotope record and Late Quaternary sedimentation rates at the Antarctic continental margin. In Bleil, U. & Thiede, H. (eds.), Geological History of the Polar Oceans: Arctic versus Antarctic, *NATO ASI Series C*, **308**: pp. 539-572; Kluwer Academic Publishers (Dordrecht).

Hillenbrand C.-D. & Cortese G. 2006. Polar stratification: A critical view from the Southern Ocean. *Palaeogeography, Palaeoclimatology, Palaeoecology*, **242** (3-4): 240-252.

Hillenbrand C.D., Crowhurst S.J., Williams M., Hodell D.A., McCave I.N., Ehrmann W., Xuan C., Piotrowski A.M., Hernández-Molina F.J., Graham A.G.C., Grobe H., Williams T.J., Horrocks J.R., Allen C.S. & Larter R.D. 2021. New insights from multi-proxy data from the West Antarctic continental rise: Implications for dating and interpreting Late Quaternary palaeoenvironmental records. *Quaternary Science Reviews*, **257**: 106842.

Hillenbrand C.D., Kuhn G. & Frederichs T. 2009. Record of a Mid-Pleistocene depositional anomaly in West Antarctic continental margin sediments: an indicator for ice-sheet collapse? *Quaternary Science Reviews*, **28** (13-14): 1147-1159.

Hillenbrand C.D., Moreton S.G., Caburlotto A., Pudsey C.J., Lucchi R.G., Smellie J.L., Benetti S., Grobe H., Hunt J.B. & Larter R.D. 2008. Volcanic time-markers for Marine Isotopic Stages 6 and 5 in Southern Ocean sediments and Antarctic ice cores: implications for tephra correlations between palaeoclimatic records. *Quaternary Science Reviews*, **27** (5-6): 518-540.

Jaccard S.L., Hayes C.T., Martinez-Garcia A., Hodell D.A., Anderson R.F., Sigman D.M. & Haug G.H. 2013. Two Modes of Change in Southern Ocean Productivity Over the Past Million Years. *Science*, **339**: 1419-1423.

Jimenez-Espejo F.J., Presti M., Kuhn G., McKay R., Crosta X., Escutia C., Lucchi R.G., Tolotti R., Yoshimura T., Ortega Huertas M., Macrì P., Caburlotto A. & De Santis L. 2020. Late Pleistocene oceanographic and depositional variations along the Wilkes Land margin (East Antarctica) reconstructed with geochemical proxies in deep-sea sediments. *Global and Planetary Change*, **184**: 103045.

Jones J., Kohfeld K., Bostock H., Crosta X., Liston M., Dunbar G., Chase Z., Leventer A., Anderson H. & Jacobsen G. 2021, *in review*. Sea Ice Changes in the Southwest Pacific Sector of the Southern Ocean During the Last 140,000 Years. *Climate of the Past Discussions [preprint]*.

Kim S., Lee J.I., McKay R.M., Yoo K.-C., Bak Y.-S., Lee M.K., Roh Y.H., Yoon H.I., Moon H.S. & Hyun C.-U. 2020a. Late pleistocene paleoceanographic changes in the Ross Sea – Glacial-interglacial variations in paleoproductivity, nutrient utilization, and deep-water formation. *Quaternary Science Reviews*, **239**: 106356.

Kim S., Yoo K.-C., Lee J.I., Lee M.K., Kim K., Yoon H.I. & Moon H.S. 2018. Relationship between magnetic susceptibility and sediment grain size since the last glacial period in the Southern Ocean off the northern Antarctic Peninsula – Linkages between the cryosphere and atmospheric circulation. *Palaeogeography, Palaeoclimatology, Palaeoecology*, **505**: 359-370.

Kim S., Yoo K.-C., Lee J.I., Roh Y.H., Bak Y.-S., Um I.-K., Lee M.K. & Yoon H.I. 2020b. Paleoceanographic changes in the Southern Ocean off Elephant Island since the last glacial period: Links between surface water productivity, nutrient utilization, bottom water currents, and ice-rafted debris. *Quaternary Science Reviews*, **249**: 106563.

Konfirst M.A., Scherer R.P., Hillenbrand C.D. & Kuhn G. 2012. A marine diatom record from the Amundsen Sea - Insights into oceanographic and climatic response to the Mid-Pleistocene Transition in the West Antarctic sector of the Southern Ocean. *Marine Micropaleontology*, **92-93**: 40-51.

Lee J.I., Yoon H.I., Yoo K.-C., Lim H.S., Lee Y.I., Kim D., Bak Y.-S. & Itaki T. 2012. Late Quaternary glacial–interglacial variations in sediment supply in the southern Drake Passage. *Quaternary Research*, **78** (1): 119-129.

Lisiecki L.E. & Raymo M.E. 2005. A Pliocene-Pleistocene stack of 57 globally distributed benthic δ18O records. *Paleoceanography*, **20** (1): PA1003.

Lucchi R.G., Rebesco M., Camerlenghi A., Busetti M., Tomadin L., Villa G., Persico D., Morigi C., Bonci M.C. & Giorgetti G. 2002. Mid-late Pleistocene glacimarine sedimentary processes of a high-latitude, deep-sea sediment drift (Antarctic Peninsula Pacific margin). *Marine Geology*, **189** (3-4): 343-370.

Macrì P., Sagnotti L., Lucchi R.G. & Rebesco M. 2006. A stacked record of relative geomagnetic paleointensity for the past 270 kyr from the western continental rise of the Antarctic Peninsula. *Earth and Planetary Science Letters*, **252** (1-2): 162-179.

Mazaud A., Michel E., Dewilde F. & Turon J.L. 2010. Variations of the Antarctic Circumpolar Current intensity during the past 500 ka. *Geochemistry, Geophysics, Geosystems*, **11** (8): n/a-n/a.

Menviel, L., Timmermann, A., Timm, O.E. & Mouchet, A. 2010. Climate and biogeochemical response to a rapid melting of the West Antarctic Ice Sheet during interglacials and implications for future climate. *Paleoceanography*, **25**, PA4231. https://doi.org/10.1029/2009PA001892

Mulvaney R., Abram N.J., Hindmarsh R.C., Arrowsmith C., Fleet L., Triest J., Sime L.C., Alemany O. & Foord S. 2012. Recent Antarctic Peninsula warming relative to Holocene climate and ice-shelf history. *Nature*, **489** (7414): 141-144.

Nair A., Mohan R., Crosta X., Manoj M.C., Thamban M. & Marieu V. 2019. Southern Ocean sea ice and frontal changes during the Late Quaternary and their linkages to Asian summer monsoon. *Quaternary Science Reviews*, **213**: 93-104.

Narcisi B., Petit J.-R. & Tiepolo M. 2006. A volcanic marker (92 ka) for dating deep east Antarctic ice cores. *Quaternary Science Reviews*, **25**: 2682-2687.

Nielsen S.H.H., Koç N. & Crosta X. 2004. Holocene climate in the Atlantic sector of the Southern Ocean: Controlled by insolation or oceanic circulation? *Geology*, **32** (4): 317.

Nürnberg C.C., Bohrmann G., Schlüter M. & Frank M. 1997. Barium accumulation in the Atlantic sector of the Southern Ocean: Results From 190,000-year records. *Paleoceanography*, **12** (4): 594-603.

Orme L.C., Crosta X., Miettinen A., Divine D.V., Husum K., Isaksson E., Wacker L., Mohan R., Ther O. & Ikehara M. 2020. Sea surface temperature in the Indian sector of the Southern Ocean over the Late Glacial and Holocene. *Climate of the Past*, **16** (4): 1451-1467.

Parrenin F., Barnola J.-M., Beer J., Blunier T., Castellano E., Chappellaz J., Dreyfus G., Fischer H., Fujita S., Jouzel J., Kawamura K., Lemieux-Dudon B., Loulergue L., Masson-Delmotte V., Narcisi B., Petit J.-R., Raisbeck G., Raynaud D., Ruth U., Schwander J., Severi M., Spahni R., Steffensen J.P., Svensson A., Udisti R., Waelbroeck C. & Wolff E. 2007. The EDC3 chronology for the EPICA Dome C ice core. *Climate of the Past*, **3**: 485-497.

Petit J.-R., Mounier L., Jouzel J., Korotkevich Y.S., Kotlyakov V.I. & Lorius C. 1990. Palaeoclimatological and chronological implications of the Vostok core dust record. *Nature*, **343**: 56-58.

Pudsey C.J. 2000. Sedimentation on the continental rise west of the Antarctic Peninsula over the last three glacial cycles. *Marine Geology*, **167**: 313-338.

Pudsey C.J. & Camerlenghi A. 1998. Glacial–interglacial deposition on a sediment drift on the Pacific margin of the Antarctic Peninsula. *Antarctic Science*, **10** (3): 286-308.

Pudsey C.J. & Howe J.A. 1998. Quaternary history of the Antarctic Circumpolar Current: evidence from the Scotia Sea. *Marine Geology*, **148**: 83-112.

Pugh R.S., McCave I.N., Hillenbrand C.D. & Kuhn G. 2009. Circum-Antarctic age modelling of Quaternary marine cores under the Antarctic Circumpolar Current: Ice-core dust–magnetic correlation. *Earth and Planetary Science Letters*, **284** (1-2): 113-123.

Romero O.E., Armand L.K., Crosta X. & Pichon J.J. 2005. The biogeography of major diatom taxa in Southern Ocean surface sediments: 3. Tropical/Subtropical species. *Palaeogeography, Palaeoclimatology, Palaeoecology*, **223** (1-2): 49-65.

Schweitzer P.N. 1995. Monthly average polar sea-ice concentration 1978 through 1991. *U.S. Geological Survey, Reston, Virginia*.

Shemesh A., Hodell D., Crosta X., Kanfoush S., Charles C. & Guilderson T. 2002. Sequence of events during the last deglaciation in Southern Ocean sediments and Antarctic ice cores. *Paleoceanography*, **17** (4): 8-1-8-7.

Shin J.Y., Kim S., Zhao X., Yoo K.-C., Yu Y., Lee J.I., Lee M.K. & Yoon H.I. 2020. Particle-size dependent magnetic properties of Scotia Sea sediments since the Last Glacial Maximum: Glacial ice-sheet discharge controlling magnetic proxies. *Palaeogeography, Palaeoclimatology, Palaeoecology*, **557**: 109906.

Thamban M., Naik S.S., Mohan R., Rajakumar A., Basavaiah N., D'Souza W., Kerkar S., Subramaniam M.M., Sudhakar M. & Pandey P.C. 2005. Changes in the source and transport mechanism of terrigenous input to the Indian sector of Southern Ocean during the late Quaternary and its palaeoceanographic implications. *Journal of Earth System Science*, **114** (5): 443-452.

Warnock J.P., Scherer R.P. & Konfirst M.A. 2015. A record of Pleistocene diatom preservation from the Amundsen Sea, West Antarctica with possible implications on silica leakage. *Marine Micropaleontology*, **117**: 40-46.

Weber, M.E., Clark, P.U., Kuhn, G., Timmermann, A., Sprenk, D., Gladstone, R., et al. 2014. Millennial-scale variability in Antarctic ice-sheet discharge during the last deglaciation. *Nature*, **510** (7503): 134-138.

Weber M.E., Kuhn G., Sprenk D., Rolf C., Ohlwein C. & Ricken W. 2012. Dust transport from Patagonia to Antarctica – A new stratigraphic approach from the Scotia Sea and its implications for the last glacial cycle. *Quaternary Science Reviews*, **36**: 177-188.

Weber M.E., Niessen F., Kuhn G. & Wiedicke M. 1997. Calibration and application of marine sedimentary physical properties using a multi-sensor core logger. *Marine Geology*, **136**: 151-172.

Weltje G.J. & Tjallingii R. 2008. Calibration of XRF core scanners for quantitative geochemical logging of sediment cores: Theory and application. *Earth and Planetary Science Letters*, **274** (3-4): 423-438.

Wilch T.I., McIntosh W.C. & Dunbar R.B. 1999. Late Quaternary volcanic activity in Marie Byrd Land: Potential [40]Ar/[39]Ar-dated time horizons in West Antarctic ice and marine cores. *GSA Bulletin*, **111** (10): 1563-1580.

Wu L., Wilson D.J., Wang R., Passchier S., Krijgsman W., Yu X., Wen T., Xiao W. & Liu Z. 2021. Late Quaternary dynamics of the Lambert Glacier-Amery Ice Shelf system, East Antarctica. *Quaternary Science Reviews*, **252**: 106738.

Xiao W., Esper O. & Gersonde R. 2016a. Last Glacial - Holocene climate variability in the Atlantic sector of the Southern Ocean. *Quaternary Science Reviews*, **135**: 115-137.

Xiao W., Frederichs T., Gersonde R., Kuhn G., Esper O. & Zhang X. 2016b. Constraining the dating of late Quaternary marine sediment records from the Scotia Sea (Southern Ocean). *Quaternary Geochronology*, **31**: 97-118.

Yamazaki T. & Ikehara M. 2012. Origin of magnetic mineral concentration variation in the Southern Ocean. *Paleoceanography*, **27** (2): n/a-n/a.

Yoon H.I., Yoo K.-C., Bak Y.-S., Lee Y.I. & Lee J.I. 2009. Core-based reconstruction of paleoenvironmental conditions in the southern Drake Passage (West Antarctica) over the last 150 ka. *Geo-Marine Letters*, **29** (5): 309-320.

Zielinski U. 1993. *Quantitative estimation of palaeoenvironmental parameters of the Antarctic Surface Water in the Late Quaternary using transfer functions with diatoms*. Reports on Polar Research.

Zielinski U., Bianchi C., Gersonde R. & Kunz-Pirrung M. 2002. Last occurrence datums of the diatoms *Rouxia leventerae* and *Rouxia constricta*: indicators for marine isotope stages 6 and 8 in Southern Ocean sediments. *Marine Micropaleontology*, **46**: 127-137.